# Comprehensive characterization of *PTEN* mutational profile in a series of 34,129 colorectal cancers

Ilya G. Serebriiskii [1,2✉], Valery Pavlov[2,3], Rossella Tricarico[1,4], Grigorii Andrianov[1,2], Emmanuelle Nicolas[1], Mitchell I. Parker[1,5], Justin Newberg[6], Garrett Frampton [6], Joshua E. Meyer [1,7] & Erica A. Golemis [1,8✉]

Loss of expression or activity of the tumor suppressor *PTEN* acts similarly to an activating mutation in the oncogene *PIK3CA* in elevating intracellular levels of phosphatidylinositol (3,4,5)-trisphosphate (PIP3), inducing signaling by AKT and other pro-tumorigenic signaling proteins. Here, we analyze sequence data for 34,129 colorectal cancer (CRC) patients, capturing 3,434 *PTEN* mutations. We identify specific patterns of *PTEN* mutation associated with microsatellite stability/instability (MSS/MSI), tumor mutational burden (TMB), patient age, and tumor location. Within groups separated by MSS/MSI status, this identifies distinct profiles of nucleotide hotspots, and suggests differing profiles of protein-damaging effects of mutations. Moreover, discrete categories of *PTEN* mutations display non-identical patterns of co-occurrence with mutations in other genes important in CRC pathogenesis, including *KRAS*, *APC*, *TP53*, and *PIK3CA*. These data provide context for clinical targeting of proteins upstream and downstream of *PTEN* in distinct CRC cohorts.

[1] Program in Molecular Therapeutics, Fox Chase Cancer Center, Philadelphia, PA 19111, USA. [2] Kazan Federal University, Russian Federation, 420000 Kazan, Russia. [3] Moscow Institute of Physics and Technology, Russian Federation, 141701 Dolgoprudny, Moscow Region, Russia. [4] Department of Biology and Biotechnology, University of Pavia, 27100 Pavia, Italy. [5] Molecular & Cell Biology & Genetics (MCBG) Program, Drexel University College of Medicine, 19102 Philadelphia, PA, USA. [6] Foundation Medicine Inc, 150 Second St., Cambridge, MA 02141, USA. [7] Department of Radiation Oncology, Fox Chase Cancer Center, Philadelphia, PA 19111, USA. [8] Department of Cancer and Cellular Biology, Lewis Katz School of Medicine at Temple University, Philadelphia, PA 19140, USA. ✉email: Ilya.Serebriiskii@fccc.edu; Erica.Golemis@fccc.edu

In 2019, there were estimated to be over 145,000 cases of colorectal cancer (CRC), and over 51,000 deaths, making it the third most common cause of cancer incidence and mortality in the United States for both sexes[1]. Overall survival at 5 years in patients diagnosed with the distant disease remains at 14%[1], motivating efforts to improve therapeutic options by better understanding CRC biology. Over the past two decades, it has been recognized that distinct subsets of CRCs present with different pathological features and prognoses, and respond differently to targeted therapies and radiation[2,3]. Clinically important distinguishing features for CRC include tumor subsite (e.g., colon versus rectum[4,5]); microsatellite stable (MSS) status, versus a high level of microsatellite instability (MSI-H)[6]; and presence or absence of a CpG island methylator phenotype (CIMP)[7].

As next-generation sequencing (NGS) has become a common feature of clinical management, there have been growing efforts to identify specific mutational signatures that segment CRC patients into clinically useful predictive and prognostic categories, and align molecular profiles with clinical categories such as MSI-H/MSS and tumor sub-site[2]. In some cases, this is unequivocally useful; for example, in CRC, the choice of first-line therapy depends on the presence or absence of specific mutations in *KRAS* that confer resistance to the EGFR-targeted monoclonal antibody cetuximab[8]. For *KRAS* and other genes commonly mutated in CRC (*APC*, *TP53*, *MLH1*, and *MSH2*), the significance of the presence or absence of a mutation, and in some cases, the specific clinical characteristics associated with commonly recurring mutation hotspots[9], are becoming well-understood and can help refine clinical management strategies. However, some genes that function as important tumor suppressors or oncogenes in other tumor types are mutated at a relatively low frequency in CRC, limiting assessment of their mutation patterns in this disease.

*PTEN* (phosphatase and tensin homolog deleted on chromosome ten), a tumor suppressor located at 10q23, is commonly epigenetically downregulated or somatically mutated in many types of cancer; further, germline mutations in *PTEN* are associated with PTEN hamartoma tumor syndrome (PHTS), and predisposing for some forms of cancer[10–15]. The primary biological function of PTEN is to hydrolyze phosphatidylinositol (3,4,5)-trisphosphate (PIP3) to phosphatidylinositol (4,5)-bisphosphate (PIP2), reversing a PIP2 to PIP3 conversion catalyzed by PI3K. PIP3 is required for the activity of AKT, a critical regulator of proliferative and survival signaling; elimination of *PTEN* in tumors strongly promotes AKT activity[16,17]. In addition, in some tumor types loss of PTEN activity has been shown to contribute to aggressive tumor growth in other ways, increasing cancer cell migration and invasion[18] and contributing to genomic instability, among other roles[10,11].

A number of studies have now indicated that specific *PTEN* mutations have different effects on the tumor suppressor activity of this protein; for example, minor differences in PTEN protein expression associated with distinct germline mutations can result in a significantly different impact on risk for cancer versus other diseases[12,19–21]. Hence, recognizing patterns of *PTEN* mutation is important in terms of assessing prognostic significance. In tumors such as glioblastoma or endometrial cancers, the *PTEN* gene is somatically mutated in 30–40% of tumors, and deleted in as many as 78% of tumors, making it easy to align mutations with clinical features. In contrast, somatic mutation of the *PTEN* gene has been described as occurring in fewer than 10% of CRCs[11,13]. This relatively low frequency has hindered the identification of clinically relevant patterns of *PTEN* mutation in CRC.

In this study, we analyze *PTEN* mutational patterns in a dataset of 34,129 CRC tumors from patients profiled by Foundation Medicine Inc. (FMI). This analysis, which captures data on 3434 somatic *PTEN* alterations identified in tumors, allows us to assign specific patterns of mutations as a consequence of tumor subsite, age, sex, MSI-H/MSS status, tumor mutation burden (TMB), and co-segregation with other driver mutations. These data also identify previously unreported hotspots in *PTEN*, as well as patterns of mutation affecting PTEN lipid phosphatase activity and stability, that distinguish discrete patient cohorts.

## Results

**Patient population: age, gender, tumor site, and MSI status.** We analyzed data for 34,129 colorectal (CRC) tumors profiled by NGS in the course of routine clinical care for patients with advanced disease (Table 1, Fig. 1a, Supplementary Fig. 1 and Supplementary Data 1). This cohort is comparable in clinical features to that reported in earlier studies, including a 45:55 ratio of female to male patients, a typical age distribution at the time of sequencing (average 57–59 years old, Supplementary Table 1), and an 84:16 ratio of the colon to rectal cancers. Besides comprehensive genomic profiling (CGP) for mutations in 315 cancer-related genes, this analysis established TMB (a measure of the total amount of somatic coding mutations in a tumor) and status of tumors as MSS or with high MSI-H for most of the specimens. Age generally did not affect TMB distribution for the MSS and MSI-H cohorts (Supplementary Fig. 1c).

For 30,885 of the 34,129 sequenced tumors, 1443 were clinically annotated as MSI-H, and 29442 as MSS; additional tumors were assigned as MSI-H versus MSS based on TMB as in[22] (rules defined in Fig. 1b, Supplementary Fig. 1d), resulting in two cohorts subsequently referred to as MT-L (MSS plus TMB-Low, 32,233 cases), and MT-H (MSI-H plus TMB-High, 1603 cases). Among the specimens with status defined as MSS, 243 had high TMB; this subset, designated MSS-htmb, was considered separately in some analyses. Typically, MSS-htmb tumors occurred in younger patients, while MT-H tumors were more frequent in the oldest patients (median ages 55, 59, and 64 for MSS-htmb, MT-L, and MT-H, correspondingly; Fig. 1c, Supplementary Table 1). The MT-L and MSS-htmb cohorts were both significantly biased toward males (Fig. 1d, Supplementary Table 2). Among the MT-H patients, a complicated imbalance in sex ratios was observed, with a bias toward males <60 years of age, but toward females in patients >60 years of age (Supplementary Fig. 1e). Male sex was associated with very high TMB, particularly in the MSS-htmb cohort (Supplementary Fig. 1f). Analysis by tumor location revealed a lower fraction of rectal

**Table 1 Clinical characteristics of 34,129 colorectal cancer patients in the study.**

| Site | Number | % |
|---|---|---|
| Colon | 28,582 | 83.75 |
| Rectum | 5,547 | 16.25 |
| *Sex* | | |
| F | 15,308 | 44.89 |
| M | 18,799 | 55.11 |
| *Microsatellite status* | | |
| MSI-H | 1443 | 4.2 |
| MSS | 29,442 | 86.3 |
| Unknown/ambiguous | 3244 | 9.5 |
| *Age* | | |
| mean | 59.58 | |
| sd | 12.87 | |
| median | 59 | |

*MSS* microsatellite stable, *MSI-H* microsatellite instability-high. Individuals with rectal cancer were slightly younger than colon cancer patients (Supplementary Fig. 1a); there was no age difference related to sex (Supplementary Fig. 1b).

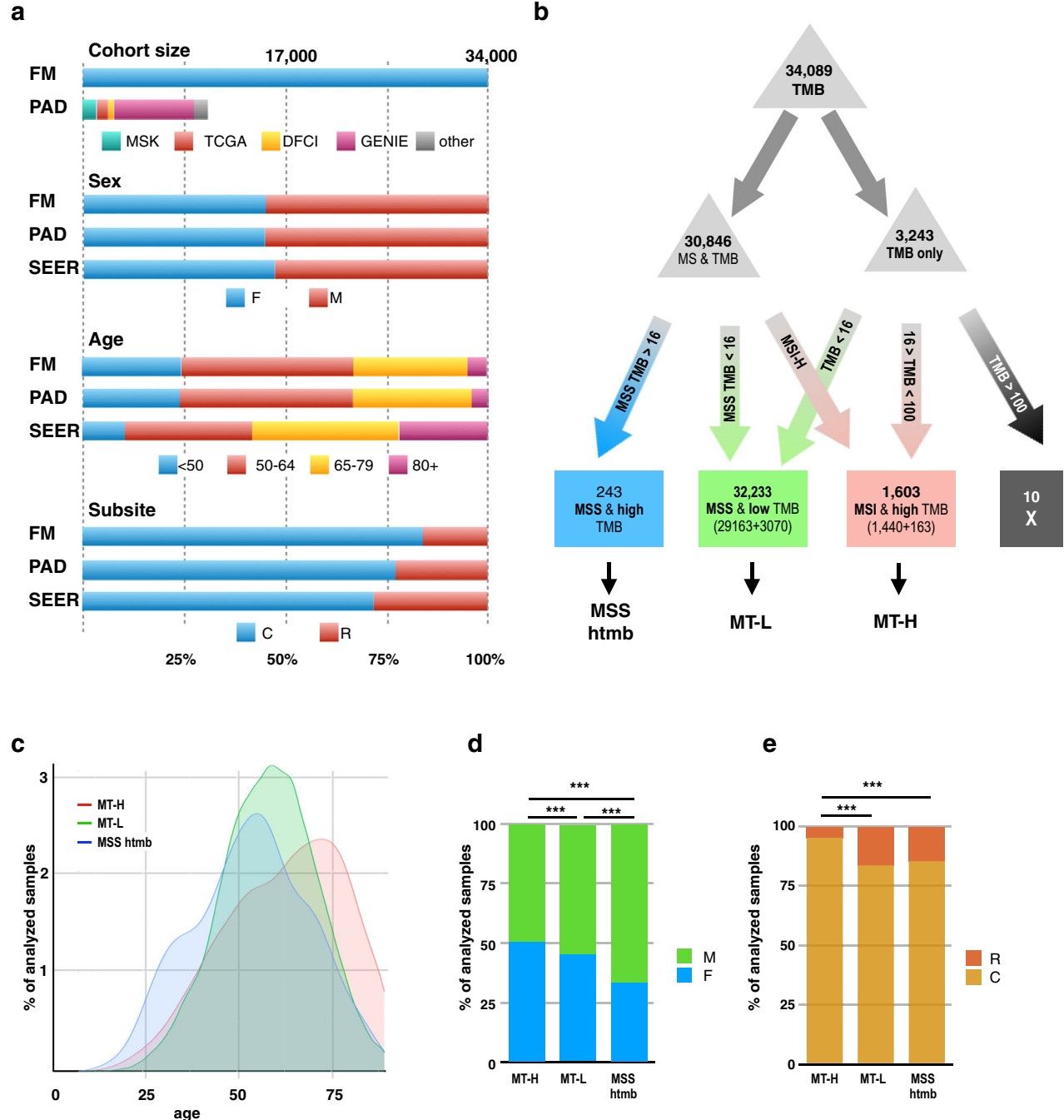

cancers among the MT-H tumors, compared to the MT-L and MSS-htmb cohorts (Fig. 1e, Supplementary Table 3).

**Overall *PTEN* mutation frequency in MT-H, MT-L, and MSS-htmb CRC.** Nonsynonymous *PTEN* mutations or *PTEN* deletions were identified in 2966 (8.7%) of the CRC specimens analyzed, comparable to previously reported frequencies of 8.1% reported by TCGA (Fig. 2a). In further concordance with TCGA, PTEN alterations are typically mutations causing amino acid changes or loss of the protein (including homozygous deletion, frameshift, nonsense, splice site, or missense mutations, or short in-frame deletions or insertions); only a single case of *PTEN* amplification and 38 rearrangements were observed in the full cohort.

The *PTEN* mutation frequency was lowest in the MT-L cohort, and higher in the MT-H cohort, matching earlier studies[23,24]; and extremely high in the MSS-htmb subset. In most cases of MT-L

CRC, most tumors contained unique *PTEN* mutations (2301/32,233 tumors; 7.2%), with only 137/32,233 (0.4%) having multiple mutations. MT-H (400/1603; 25.1%) and MSS-htmb (110/243; 45.3%) tumors were more likely to bear mutated *PTEN*, and a higher proportion of these tumors had multiple *PTEN* mutations (189/1603 (11.8%) of MT-H patients and 67/243 (27.6%) of MSS-htmb patients) (Fig. 2b, Supplementary Table 4). These differences in frequency did not passively reflect overall TMB in these tumor classes (Supplementary Fig. 2a). Although higher levels of TMB were associated with some elevation in *PTEN* mutation frequency (Fig. 2c), the degree of correlation differed between the MT-L, MT-H, and MSS-htmb sub-classes, and in no case exactly paralleled TMB.

Interestingly, the prevalence of *PTEN* mutations is higher in CRC tumors from females than from males in the MT-L subset, but higher in males than females in the MSS-htmb subset

**Fig. 1 Overall characterization of the dataset. a** Comparison of FMI dataset in the present study versus a benchmark group of publicly available data (PAD) for colorectal cancer (CRC) published by Memorial Sloan-Kettering (MSK)[94], the Dana Farber Cancer Institute (DFCI)[95], the Genomics Evidence Neoplasia Information Exchange (GENIE)[96], and The Cancer Genome Atlas (TCGA)[97]. Population characteristics are also compared to the overall population reported in SEER (Surveillance, Epidemiology, and End Results)[98]; contents accessed 5.5.2020. **b** Flowchart and analysis tree for populations defined by FMI as having microsatellite instability (MSI-H) or being microsatellite stable (MSS), and/or with known tumor mutation burden (TMB) (see also **c**, **d**). TMB cutpoints of >16 and <100 were used to generate MSI-H/high TMB (MT-H), MSS/low TMB (MT-L), and MSS/high TMB (MSS-htmb) analysis cohorts. Briefly, we previously determined that a TMB = 16 mutations/Mb segregated MSI-H tumors (TMB ≥ 16) from MSS tumors (TMB < 16) in 99% of cases[22]; similarly, in this dataset (Supplementary Fig. 1d), 98% of MSI-H tumors are above this threshold, and 99% of MSS tumors below this threshold. Hence, using this metric to segregate the remaining 3243 of 3244 tumors for which only TMB was available, all specimens with TMB < 16 were grouped with MSS tumors, resulting in 32,233 CRC tumors designated MT-L (MSS plus TMB-Low). Among tumors with defined MSI-H status, ~95% had TMB < 100; however, among tumors with a very high TMB (>100), there were comparable numbers of MSI-H and MSS tumors (Supplementary Fig. 1c, left panel). Hence, among tumors where only TMB was known, those with TMB ≥ 16, but <100 were assigned as MT-H (MSI-H plus TMB-High), and those with TMB >100 were not considered further. Graphics design of panels **a**, **b** is slightly modified from[22], reporting a smaller dataset. **c** Age distribution of patients with CRC designated as MT-H (pink), MT-L (green), or MSS-htmb (blue). **d**, **e** Composition of the MT-H, MT-L, and MSS-htmb groups by (**d**) sex or (**e**) colon (C) versus rectum (R) tumor subsite. *** indicate $p < 0.001$. Sample sizes: MT-H – 1600; MT-L – 32212; MSS htmb—242. Calculated sex and subsite fractions, as well as $p$-values for the comparisons between subsets (calculated using the two-sample test for equality of proportions with continuity correction) are provided in Supplementary Tables 3 and 4. Summary level data for the FMI CRC dataset are provided as Supplementary Data 1.

($p = 2 \times 10^{-17}$ and 0.03, respectively); there was no significant difference for *PTEN* mutation prevalence based on sex in the MT-H subset (Fig. 2d, Supplementary Table 5). The prevalence of *PTEN* mutations is higher in the colon than in the rectum (Fig. 2e, Supplementary Table 6), with the difference reaching statistical significance in the MT-L subset ($p = 6.1 \times 10^{-10}$). However, the impact of age differs strikingly between the three tumor subsets. In MT-L tumors, the prevalence of *PTEN* alterations significantly increases by age ($p = 1.77 \times 10^{-7}$) (Fig. 2f, Supplementary Table 7), at similar rates in the colon and rectum subsites, and in males and females (Supplementary Fig. 2b, c and Supplementary Tables 8 and 9). In MT-H tumors, the overall increase of prevalence of *PTEN* alterations by age did not reach statistical significance, and did not vary by sex; but there were markedly different age trends by subsite (Supplementary Fig. 2d, e and Supplementary Tables 7–9). Conversely, while age is associated with a decrease in *PTEN* mutation frequency in MSS-htmb tumors (Fig. 2f, $p = 0.001$), there was no difference in age trends based on subsite or sex (Supplementary Fig. 2f, g and Supplementary Tables 7–9).

***PTEN* mutation class based on MS status, sex, tumor subsite, and age**. Besides differences in mutation frequency, there were significant differences in the categories of mutation occurring in different tumor subtypes (Fig. 2g, Supplementary Table 10). In MT-L tumors, large homozygous deletions predominated, representing 41% of all alterations; truncating mutations (33%) and potentially less damaging missense and small in-frame indels (25%) were also common. In contrast, only a single deletion was found among the MT-H tumors, with 70% of detected mutations truncating PTEN, and the remaining 30% missense/indels. These patterns reflect the well-defined mutual exclusivity of chromosomal instability and MSI in CRC, which pertains in all except a small subset of CRCs[25,26]. In the MSS-htmb tumors, this pattern is reversed, with 36% of truncating mutations, 62% missense/indels, and only 2% large deletions. These patterns were not affected significantly by sex, tumor subsite, or by age (Supplementary Fig. 2h–j and Supplementary Tables 11–13).

**PTEN mutation hotspots differ between MT-L, MT-H, and MSS-htmb cohorts**. The PTEN protein structure includes a short N-terminal regulatory region (the phosphatidylinositol-4,5-bisphosphate-binding domain (PBD)), a catalytic phosphatase domain (residues 14–185), a C2-domain that mediates phospholipid binding and protein localization (residues 190–350), and

a C-terminal tail (residues 351–403) that encompasses a PDZ domain-binding motif and phosphorylation sites that contribute to protein stability[10,27] (Fig. 3a, Supplementary Table 14).

Previous analyses have noted the concentration of missense mutations in the exons encoding the catalytic phosphatase domain, and of truncating mutations in the C2 domain[12,20,28–30]. A similar pattern was observed among the 2966 *PTEN* mutations in the merged CRC cohort, with missense and indel mutations most commonly located in sequences encoding the phosphatase domain, and truncating nonsense and frameshift mutations in the C2 domain (Fig. 3a). Overall, there were 54 hotspot mutations in the overall CRC dataset, of which 30 were extremely common in multiple forms of cancer, and had been previously reported, while 24 were novel (Fig. 3c, Supplementary Fig. 3a and Supplementary Table 15). Within the MT-L and MT-H cohorts, no differences in frequency of the most common hotspots were associated with sex (Supplementary Fig. 4a) or age (Supplementary Fig. 4b), but minor differences in hotspot preference differentiated the colon and rectal subsites in the MT-L cohort (Supplementary Fig. 4c).

In total, a large proportion of the total mutations were found in hotspots (~51% for MT-L, ~64% for MT-H, and ~71% for MSS-htmb (Fig. 3b, Supplementary Table 16). Applying saturation analysis[31] to the current dataset suggests that the detection of additional hotspots would require a very significant increase in cohort size, implying this analysis is approaching saturating for CRC overall (Supplementary Fig. 3e). There were marked qualitative differences in hotspot profile between the MT-L, MT-H, and MSS-htmb tumor classes that were not attributable to differences in sample size (Fig. 3b–f, Supplementary Fig. 3b–d and Supplementary Table 15). Among the identified hotspots, 3 (in codons C124, Q219, and Q298) were specific for the MT-L cohort and 3 (F341, R41, and K183) for the MSS-htmb cohort. Seven additional sites of elevated mutation were less commonly mutated (S170, Y76, N31, L146, C105, D92, and M134), but identifiable in the combined dataset (Supplementary Table 15). The hotspot pattern observed for the merged FMI CRC cohort was generally in good concordance (although more extensive) with that available for the TCGA CRC dataset, but differed from those seen in other tumor types (Supplementary Fig. 5). Of the mutations that did not occur in hotspots, there was in some cases a propensity to cluster in linear regions of the primary amino acid sequence (e.g., res 39–49, and 244–255), as residues within these areas are more commonly mutated than at random (Fig. 3c). The presence of these mutation-enriched regions also differed between CRC subclasses (Fig. 3d–f).

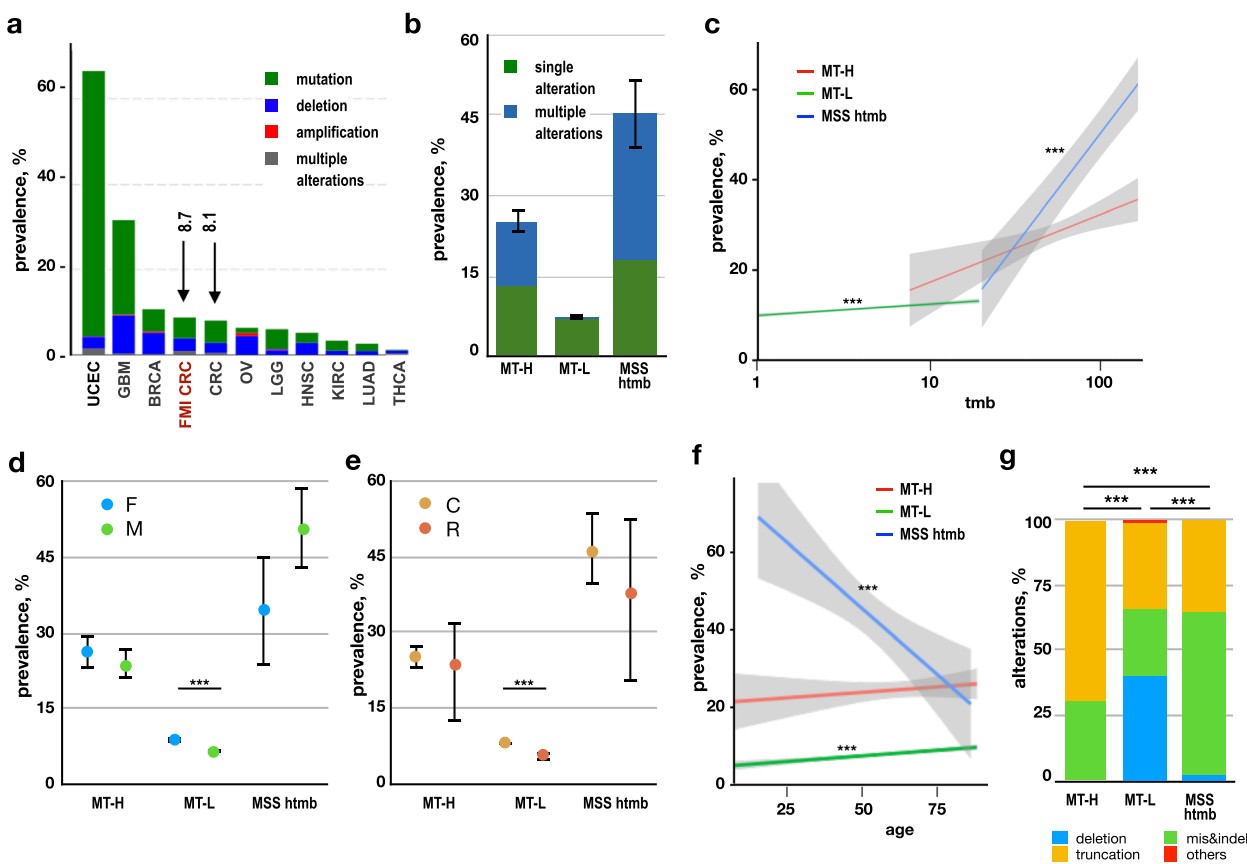

**Fig. 2 Frequency of PTEN alterations. a** Frequency of *PTEN* alterations in distinct cancer subtypes, based on analysis of the TCGA Pancancer datasets with over 500 samples accessed through cBioportal[99,100], benchmarked to data in this study (CRC-FMI). Green, mutation (missense, small indel); blue, deep deletion; red, amplification; purple, fusion; gray, multiple alterations. GBM glioblastoma multiforme, BRCA breast cancer, CRC colorectal cancer (COAD-READ, colon adenocarcinoma, and rectal adenocarcinoma), LGG low-grade glioma, KIRC clear cell renal cell carcinoma (kidney renal carcinoma), LUAD lung adenocarcinoma. **b** Frequency of tumors with *PTEN* alterations (any type) in MT-L, MT-H, and MSS-htmb tumors, indicating tumors bearing single (green) versus multiple (dark blue) mutations in *PTEN*. Sample sizes, calculated prevalence, and values of the error bars (which represent 95% confidence intervals for the prevalence of any *PTEN* alterations) are provided in Supplementary Table 4. **c** Frequency of tumors with *PTEN* alterations (any type) as a factor of TMB for MT-L (green), MT-H (red), or MSS-htmb (blue) tumors. Shaded areas represent 95% confidence intervals. *** indicates statistically significant trends (using logistic regression model), with $p = 3.07e-15$ for MT-L and $p = 2.83e-10$ for MSS htmb; $p = 0.0053$ for MT-H subset was not considered significant. **d**, **e** Frequency of tumors with PTEN mutations (any type) based on sex (panel **d**; F, female; M, male) or tumor subsite (panel **e**; C, colon; R, rectum). Error bars represent 95% confidence intervals for the estimate of the prevalence of mutations in the general population of individuals with CRC, based on the size of the current sample; relationships between PTEN mutation prevalence and patient characteristics were assessed using the two-sample test for equality of proportions with continuity correction); *** indicate $p < 0.001$. Sample sizes, calculated prevalence, and exact *p*-values are provided in Supplementary Tables 5 and 6. **f** Frequency of PTEN mutations (any type) based on age in the MT-L, MT-H, and MSS-htmb groups. Shaded areas represent 95% confidence intervals. *** indicates statistically significant trends (using a logistic regression model), with $p = 1.6E-07$ for MT-L and $p = 0.00067$ for MSS htmb; sample sizes, logistic regression coefficients, and exact *p*-values are provided in Supplementary Table 7. **g** Frequency of mutation types in MT-H, MT-L, and MSS-htmb CRC. Blue, deep deletion; green, missense, and inframe indels; gold, truncating (nonsense, splice, frameshift); red, others (including amplification and rearrangements). *** indicates statistically significant differences in types of mutation, with a *p*-value < 2.2e−16 in each case, calculated using a chi-squared contingency table test. Source data and exact proportions are provided in Supplementary Table 10. Sample sizes for panels **b**–**g**: MT-H-1587; MT-L-31,772; MSS htmb-239.

**Mutational signature profiles of MT-L, MT-H, and MSS-htmb CRC.** The non-identical pattern of *PTEN* mutations seen in the three CRC tumor sub-types, and distinct tumor sites, may reflect distinct selection pressures for discrete mutation types, differences in underlying mutational processes, or both. We first considered differences in mutational processes associated with distinct tumor mutational signatures. Among the signatures known to be common in CRC, a few could be assigned with reasonable confidence[32]. Of these, the clock-like SBS1 signature arises from the deamination of 5-methylcytosine to thymine. The SBS10a, SBS10b, and SBS28 signatures are associated with the presence of mutations impairing polymerase epsilon (POLE) exonuclease function during replication[33]. The ID1/ID2/ID5/

ID7 signatures (collectively designated hereafter as IDT, for total) are associated with gain or loss of a nucleotide in homopolymer runs (typically of As and Ts), with some signatures demonstrated to arise due to slippage during DNA replication or defective mismatch repair (MMR), and associated with MSI-H.

We aligned the nucleotide changes affecting the PTEN coding sequence with these signatures (Fig. 4a, and Supplementary Fig. 6). For the overall CRC cohort, the largest group of recurrent mutations was consistent with an SBS1 signature, reflecting 12–18% of all missense mutations in the various cohorts (Fig. 4a and Supplementary Fig 6). Mutations associated with IDT signatures were preferentially associated with the MT-H tumor subset (51% of total mutations, versus 5% in MT-L), in agreement

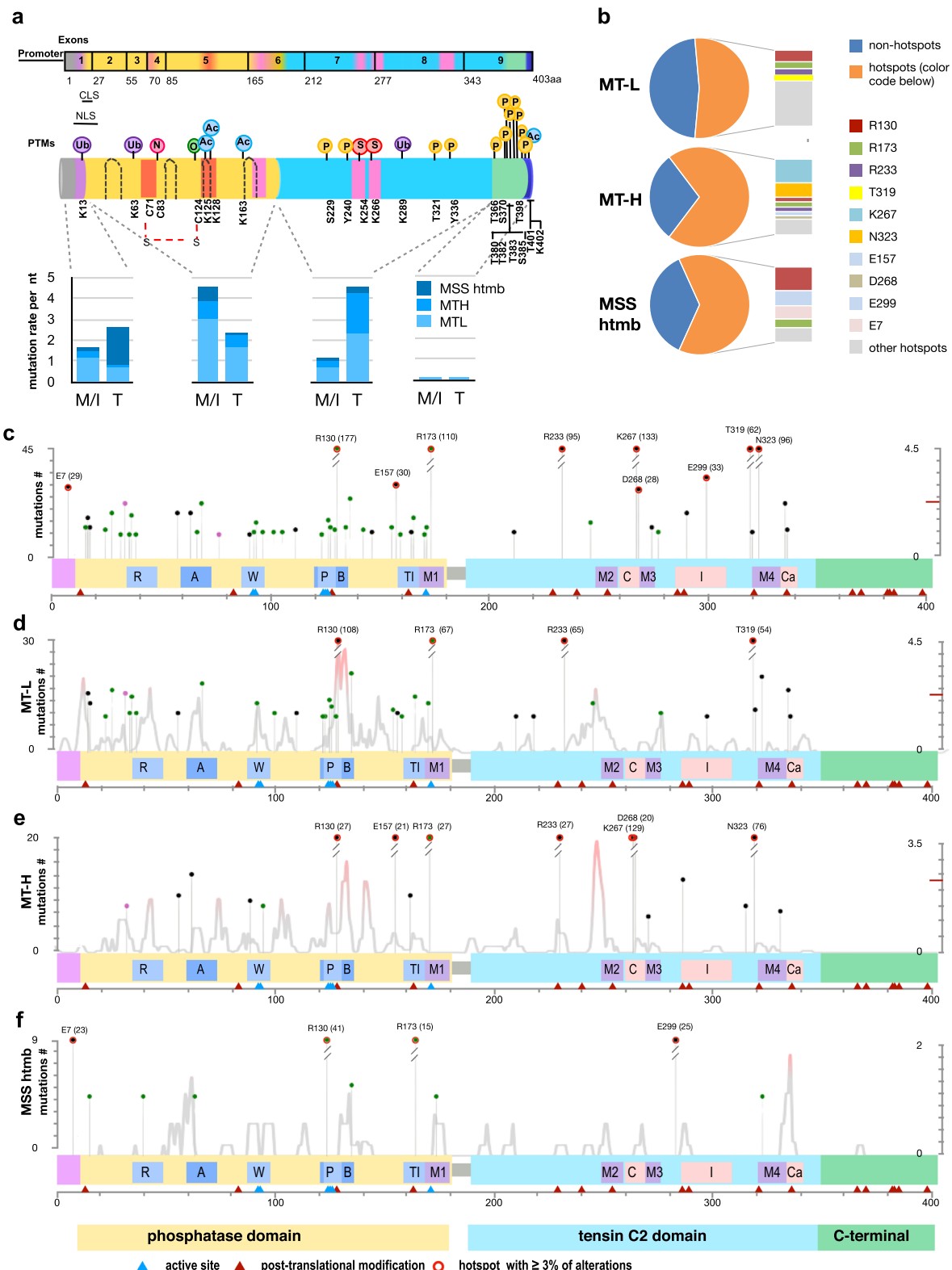

phosphatase domain | tensin C2 domain | C-terminal

▲ active site   ▲ post-translational modification   ⊙ hotspot with ≥ 3% of alterations

with previous observations[34]. SBS10a, SBS10b, and SBS28 were highly enriched in samples bearing *POLE* mutations (typically affecting exonuclease function). For the MSS-htmb subset, in which 54% of tumors are *POLE*-mutated, mutations compatible with these signatures comprise over 60% of all mutations (Fig. 4a, Supplementary Fig. 6).

Interestingly, although both the SBS1 and IDT signatures have been described as "clock-like"[32,35], accumulating as a factor of

age, an age-associated increase in these signatures among *PTEN* mutations was not observed in either the MT-L or MT-H cohorts in spite of the overall increase in *PTEN* mutations in these tumor groups (Fig. 4b and Supplementary Table 17). Combined, signatures linked to deamination, or defects in *POLE* or MMR account for the majority of mutational hotspots in the complete CRC cohort (Fig. 4c and Supplementary Table 18), and for overall mutations in the MT-H and MSS-htmb. In contrast, for MT-L

**Fig. 3 Mutation hotspots affecting the PTEN protein. a** Top, schematic of PTEN protein domain structure. Structural domains include a phosphatidylinositol 4,5-bisphosphate (PIP2)-binding domain (PBD; 6-15aa; purple), a phosphatase domain (14–185aa; yellow), C2 domain (190–350aa; light blue), a C-terminal tail (352-402aa; green) and a PDZ-binding domain (PDZ-BD; 401–403aa; blue). ATP-binding motifs (orange), intermotifs (pink), and loops (dashed lines) are also indicated. Post-translational modifications that regulate PTEN enzymatic activity are indicated (references are in Supplementary Table 14). U: ubiquitynation; N: S-nitrosylation; O: oxidation; Ac: acetylation; S: sumoylation; P: phosphorylation. Exon structure is indicated above protein. M/I, missense or inframe indel. T, truncating mutation (frameshift, nonsense). NLS: Nuclear localization sequence (8–32aa); CLS cytoplasmic localization sequence (19–25aa). Bottom, distribution of total number of mutations in the PBD, phosphatase, C2, and C-terminal domains is indicated for the MT-L, MT-H, and MSS-htmb tumors. **b** Percent of total mutations occurring at hotspot mutations (piechart), and concentration of mutations at strongly preferred amino acid hotspots (>3% of total mutations observed) for MT-L (top), MT-H (middle), and MSS-htmb (bottom) tumors. **c–f** Location of hotspots, and density of non-hotspot mutations (all classes, including truncating mutations) identified in the complete CRC cohort (**c**), or the MT-L (**d**), MT-H (**e**), or MSS-htmb (**f**) subsets. The height of each lollipop indicates the count of the corresponding mutation in the dataset (left Y-axis). Red circles on lollipops, hotspots representing >3% of total mutations observed in at least one subset. Density distribution (light gray line) represents the probability of statistically significant concentration of non-hotspot mutations along the primary structure of PTEN and is plotted as −log10(p) on the right y-axis, with the values above the indicated 2.3 threshold corresponding to p-values below 0.005. Protein features shown in **c–f** (coordinates in aa): R, Arginine loop (35-49); A, ATP-binding type-A motif (60–73); W, WPD loop (88–98); P, P loop (123–131); B, ATP-binding type-B motif (122–136); TI, TI loop (160–171); M1, Inter-domain Motif 1 (169–180); M2, Inter-domain Motif 2 (250–259); C, CBR3 loop (260–269); M3, Inter-domain Motif 3 (264–276); I, Internal loop in C2 domain (286–309); M4, Inter-domain Motif 4 (321–334); Cα, Cα2 loop (321–342). Blue triangles, active site (aa 92, 93, 124–126, 129, 130, 171); brown triangles, most common post-translational modifications as in (**a**). A number of PTEN mutations analyzed in panels (**b–f**): MT-H − 581; MT-L- 1319; MSS htmb − 203.

tumors, ~70% of mutations could be not unambiguously assigned to any specific mutational signature (Fig. 4a). Given the preponderance of MT-L tumors in the overall CRC cohort, a considerable diversity of mutations was observed that were not attributable to any specific mutational process (Fig. 4a, d). The concentration of these mutations in functionally important domains argued for the selection of mutations at the protein level, regardless of the originating source.

**Consequences of *PTEN* mutation patterns for protein structure and function**. Distinct *PTEN* mutations cause differing degrees of biological impairment depending on which PTEN protein functions they compromise[36,37]. Although the primary activity of PTEN is as a homodimeric lipid phosphatase controlling PIP3 availability, other activities including roles in protein phosphorylation, and as a non-catalytic scaffolding protein, contribute to its activity as a tumor suppressor[38,39]. Mutations that disrupt PTEN interaction with partner proteins[38], or PTEN homodimerization[40], will have differing effects on PTEN activity. Recognizing these patterns of PTEN mutation in CRC may predict the efficacy of therapies targeting PI3K, AKT, and other PTEN-associated signaling pathways[41,42]. This has led to extensive past efforts to annotate PTEN mutations for pathogenic effects on protein stability, phosphatase activity, interaction with substrates, intracellular localization, and other features[20,29,36–38,43–50].

We first analyzed the distribution of *PTEN* mutations affecting coding sequence in the CRC cohort (Figs. 3, 5a–c, Supplementary Figs. 3 and 7, Supplementary Table 19). The most damaging classes of PTEN mutations include those targeting the catalytic site of the protein, disrupting the structural integrity of either the phosphatase or the C2 domain, or truncating the protein within these domains. In the overall CRC dataset, 1148 out of 2124 total mutations resulted in the truncation of the protein prior to the C-terminal end of the C2 domain. The relative frequency of such truncating mutations differed in the MT-L, MT-H, and MSS-htmb groups, with the greatest number in the MT-H subset, arising from frameshifting small indels (Figs. 2g, 3a and Supplementary Table 19). A smaller number of truncating mutations arose from mutations in introns affecting the splicing of *PTEN* (154 cases), and rearrangements (38 of 3434 total *PTEN* alterations); these were equally represented in all tumor subtypes (Supplementary Fig. 8).

The catalytic activity of PTEN requires the integrity of a complex cleft formed by the interaction of the phosphatase and

C2 domains of PTEN (Fig. 5a). Key structural elements include the P-loop (aa 123–130), the WPD loop (aa 88–98), and the T1 loop (aa 160–171) in the phosphatase domain, and additional sequences provided by the C2 domain that help stabilize PTEN interaction with substrates[20]. Previous studies have noted the concentration of cancer-associated missense mutations around the catalytic cleft[30,51]. This pattern is also confirmed in the current study (Fig. 5b, c). Of the missense and non-frameshifting small indel mutations in the CRC analysis, ~25% (243/970) mutations overall and ~19% of the mutations in the hotspots targeted the catalytic cleft and sequences adjacent on the surface of the phosphatase and C2 domains. Although the overall rate of mutation is comparable between the phosphatase and C2 domains, there are a much higher fraction of missense/small indel mutations in the phosphatase domain in the overall CRC cohort (Fig. 3a). Of the most common of these hotspots, R130 lies in the active site pocket, and R173 at the phosphatase-C2 domain interface (Fig. 5b, c). Of the limited missense/indel hotspots distant from this interface, P246 in the C2 domain is most notable; mutations in this sequence have been suggested to interfere with the appropriate positioning of the active site toward the membrane[30].

In considering solely missense/indel hotspots in the complete CRC dataset, 42 amino acids were targeted >6 times (combining all substitutions observed at a given position (Supplementary Fig. 9 and Supplementary Table 20). The greatest density of non-truncating hotspots localized to the phosphatase domain, with peaks roughly coinciding with the R loop, ATP A binding site, and the WPD-, P-, and TI-loops (Supplementary Fig. 9). Of 42 hotspots, 32 sites were predominantly found in the MT-L subset, mostly not detectable in MT-H and MSS-htmb tumors due to much smaller numbers in these cohorts. However, over half of the hotspots identified in the MT-H and MSS htmb cohort were specific for those subsets (Supplementary Table 20). For some residues, multiple amino acid substitutions were observed, with a variance of substitution in distinct tumor subtypes (Fig. 4d, Supplementary Fig. 6). As one example, R130, located at the end of the P-loop, was the frequent site of both truncating mutations and a pathogenic missense mutation, R130Q, both associated with an SBS1 signature. Although the overall frequency of R130* and R130Q mutations did not differ between MT-H and MT-L tumors, only R130Q substitutions were present in MSS-htmb tumors, potentially reflecting the specific elevation of the SBS10b mutational signature in this group.

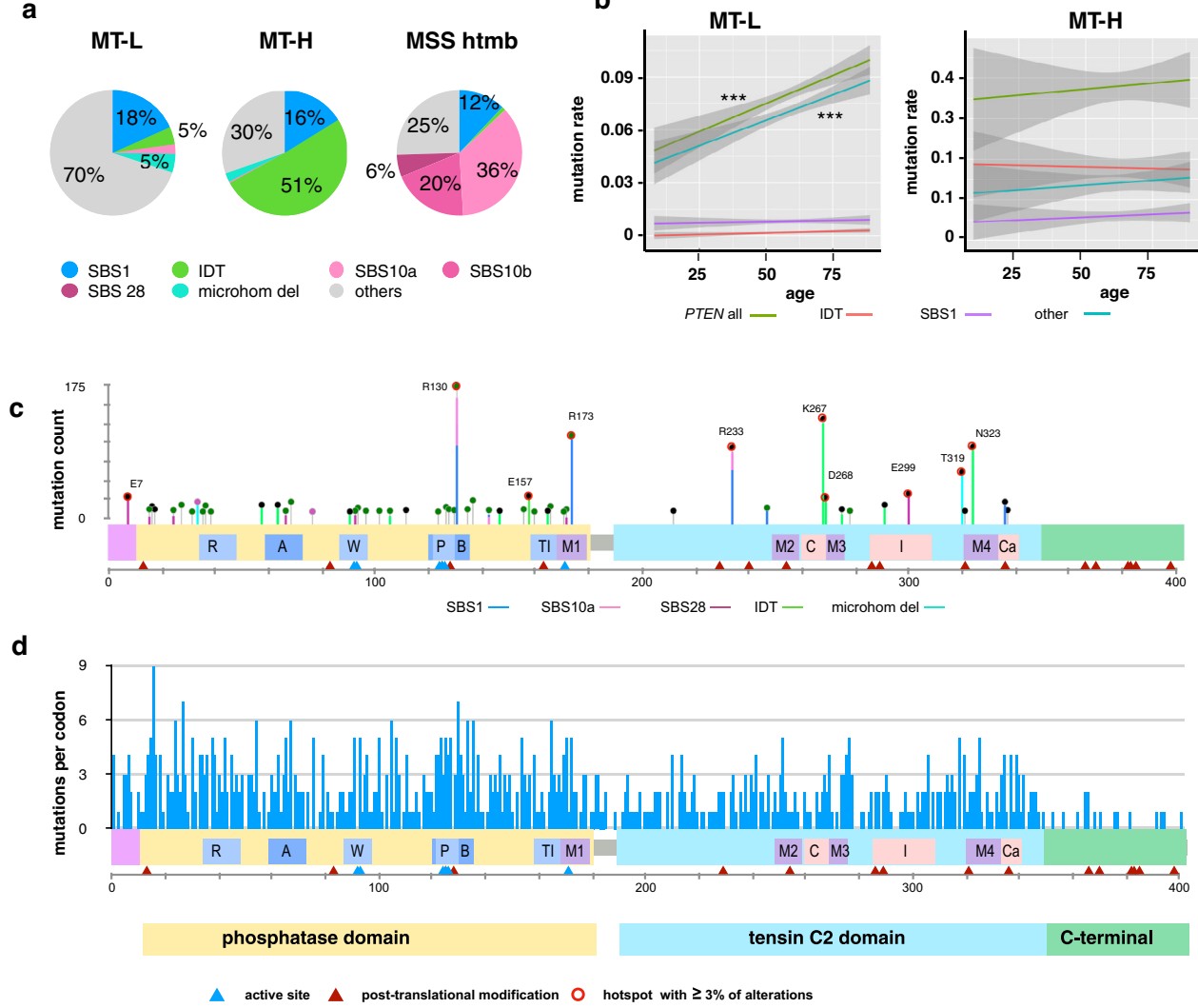

**Fig. 4 Mutation signatures associated with non-synonymous *PTEN* mutations affecting coding sequence. a** Distribution of mutational signatures across the CRC subtypes. A number of PTEN mutations were analyzed: MT-H − 606; MT-L − 1440; MSS htmb − 208. **b** Age trends for all mutations affecting *PTEN* nucleotide sequence, mutations associated with the SBS1 and IDT signatures, and mutations not defined by either SBS1 or IDT signatures (other). Shaded areas represent 95% confidence intervals. *** indicates statistically significant trends (using a generalized linear model), with $p = 7.07E−07$ for MT-L (all PTEN mutations) and $p = 1.44E−06$ for MT-L (non-SBS1, non-IDT PTEN mutations); sample sizes, regression coefficients, and exact *p*-values are provided in Supplementary Table 17. **c** Mutational signatures defining some of the hotspots; line color reflects key in (**a**). **d** Diversity of changes occurring at each codon. Bar height indicates the number of different alterations (including missense mutations, truncating mutations, or indels) arising from mutations at each indicated codon, underscoring the complexity of the mutational landscape.

Based on the analysis of the several available crystal structures of PTEN, we also identified 60 3D hotspots, defined as mutations enriched in close proximity within the tertiary folded protein structure (Supplementary Data 2). Most of these 3D hotspots cluster in the phosphatase domain (Supplementary Fig. 9), with ~24% (95/402 of the mutations in 3D hotspots) being in the catalytic cleft; an additional 3D hotspot cluster localized to the C2 domain (Supplementary Fig. 7c).

**Broader patterns of phosphatase activity and protein abundance associated with CRC *PTEN* mutations**. As an alternative approach to analyzing the consequences of *PTEN* mutations in the CRC cohort, we leveraged two published datasets probing PTEN lipid phosphatase activity (LPA) and protein abundance, in an approach similar to ref. [52]. Data from the extensive analysis of LPA in yeast[37,45] captures ~95% of the non-frameshift mutations

from the CRC cohort. Based on this analysis (Fig. 6a), 60% of missense mutations fall below the threshold of −1.1, indicating some level of impaired phosphatase activity. However, the profiles of LPA scores differ between the MT-H, MT-L, and MSS-htmb tumor subclasses, with greater loss of phosphatase activity in MT-L tumors versus MT-H and MSS-htmb tumors (Fig. 6a, *p*-values 0.0005 and 0.0006, respectively). Phosphatase impairment profiles did not differ significantly by sex, subsite, or age (Supplementary Fig. 10a, b; Supplementary Table 21).

An orthogonal dataset, assessing Variant Abundance by Massively Parallel Sequencing (VAMP-seq), established the effect of some classes of PTEN mutations on protein abundance in vivo[43,44]. This dataset provides a model for ~43% of the non-frameshift mutations from the CRC cohort. Based on VAMP-seq analyses (Fig. 6b, Supplementary Fig. 10c, d; Supplementary Table 22), and using a cut-off score of 0.4 (as in[43,44]) to indicate a

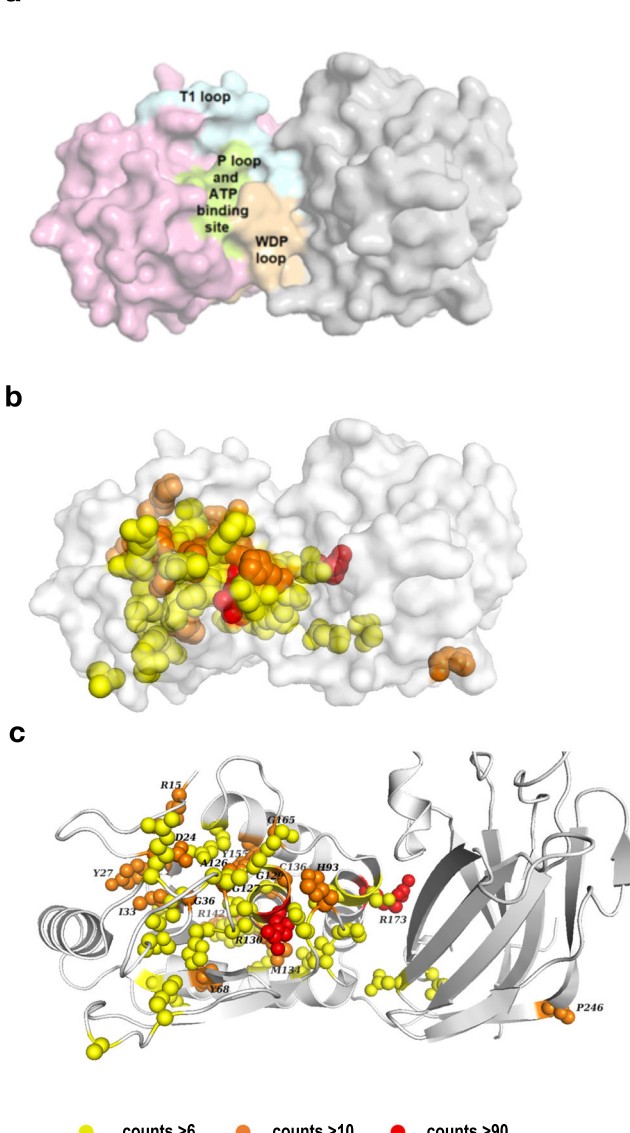

**Fig. 5 Distribution of mutations in PTEN protein domains. a** Location of the key elements of the phosphatase domain on PTEN 3D structure (modeled from pdb: 1D5R [27]). Light orange: WDP loop (aa 88–98); Limon: P loop and ATP-B binding site (123–136); Pale Cyan: TI loop (160–171); Light pink: the rest of the phosphatase domain. **b** 3D representation of the location of essential motifs for PTEN phosphatase function. **b, c** Location of missense/indel hotspots in the complete FMI CRC cohort, shown in overall structure (**b**) or zoomed into the catalytic cleft (**c**). Yellow: counts >6 (R15, D24, N31, M35, P38, R47, P95, I101, C105, H123, C124, G127, G129, T131, G132, R159, Q171, D252, and T277); Orange: counts >10 (Y27, I33, G36, Y68, H93, A126, C136, Y155, G165, and P246 (see Supplementary Table 20); Red: counts >90 (R130 and R173).

caused loss of LPA, and in many cases also reduced PTEN abundance. These estimates did not significantly vary based on tumor subsite (Supplementary Fig. 11a). However, we also performed a more nuanced analysis of the observed range of VAMP and MAVE values in light of analyses of additional properties of PTEN variants, including potential for dominant-negative (DNE) activity[53]. This revealed a more complex pattern of mutational consequences in which specific mutations altered phosphatase activity, protein abundance, both, or neither in the full CRC cohort (Fig. 6e) and distinguished the MT-L and MT-H sub (Supplementary Fig. 11b, c).

Notably, truncating mutations, which typically reduce both phosphatase activity and stability, are more common in MT-H tumors. Mutations affecting phosphatase activity but not stability are likely to possess DNE activity, based on the function of PTEN as a homodimer. DNE mutations are significantly more common in the MT-L subset than in MT-H subset, ~11% versus ~7.6% (p-value 0.0004). Among the hotspot mutations detected in the full CRC cohort, there is a particularly strong selection for DNE action and loss of function (Fig. 6f), also reflected in the reduced LPA and/or reduced protein abundance (Fig. 6g, h and Supplementary Table 23). Finally, some hotspots are comprised of mutations that retain LPA (based on annotation in the literature) but have unique effects on protein function which may impact response to targeted or chemotherapies in CRC (Supplementary Fig. 11d). Examples of these include hotspots at K66, R142, and Y336 (e.g.,[53]). Overall, these results suggest distinct functional consequences of PTEN mutations in MT-H versus MT-L tumors.

**Patterns of *PTEN* loss of heterozygosity (LOH) and multiple mutations in MT-L, MT-H, and MSS-htmb CRC.** There was also a notable non-random variation in patterns of LOH of *PTEN*, that differed by tumor subclass (Fig. 7a). LOH was much more common in the MT-L CRCs; coupled with the higher incidence of DNE mutations in the MT-L cohort, this implies a high percentage of MT-L tumors have a complete loss of PTEN function. In contrast, although the frequency of single mutations in PTEN is higher in the MT-H and MSS-htmb tumors, there is much less frequent LOH (Fig. 7a). LOH patterns did not differ based on tumor subsite or sex in the MT-L and MT-H cohorts (Supplementary Fig. 12 and Supplementary Table 24). In the MSS-htmb subset, LOH patterns differed between the colon and rectal sites (p = 0.0002); however, this difference was based on a relatively small number of samples.

Interestingly, analysis of the pattern of hotspot mutations implies some are biased to occur in the presence or absence of a wild-type *PTEN* allele, or with a second mutation in *PTEN* (Fig. 7b). Further, the number of tumors bearing multiple independent mutations in *PTEN* is significantly elevated over the expected random mutation frequency in the MT-L and MT-H subsets, while single mutations are much less frequent than expected based on random occurrence in MT-H and MSS htmb subsets (Fig. 7c). These patterns achieve higher statistical significance if only mutations predicted to compromise PTEN protein function are considered (Supplementary Table 25). There were no significant differences in TMB between subsets of samples with multiple vs single *PTEN* mutations, nor in the likelihood of acquiring an initial versus subsequent PTEN mutation (Supplementary Fig. 13, Supplementary Table 26). Based on analysis of the complete CRC cohort, some specific hotspot mutations tend to co-occur (Fig. 7d); typically, co-occurring mutations are predicted to arise from a similar mutational process based on mutational signatures linked to POLE mutations (predominant in MSS-htmb tumors), or likely

significant effect, about half (54–58%) of PTEN mutations reduce protein abundance in all CRC cohorts, without significant variation based on tumor subsite, age, or sex.

Based on simple integration of results from MAVE and VAMP cut-off values for functional impairment with reports in clinical databases and published literature (Fig. 6c), ~ 90% of mutations in the MT-L and MT-H subsets, and ~80% of the mutations in the MSS-htmb subset are predicted to have a partial or complete loss of function (Fig. 6d, Supplementary Data 3). This includes almost all of the hotspot mutations, which almost uniformly

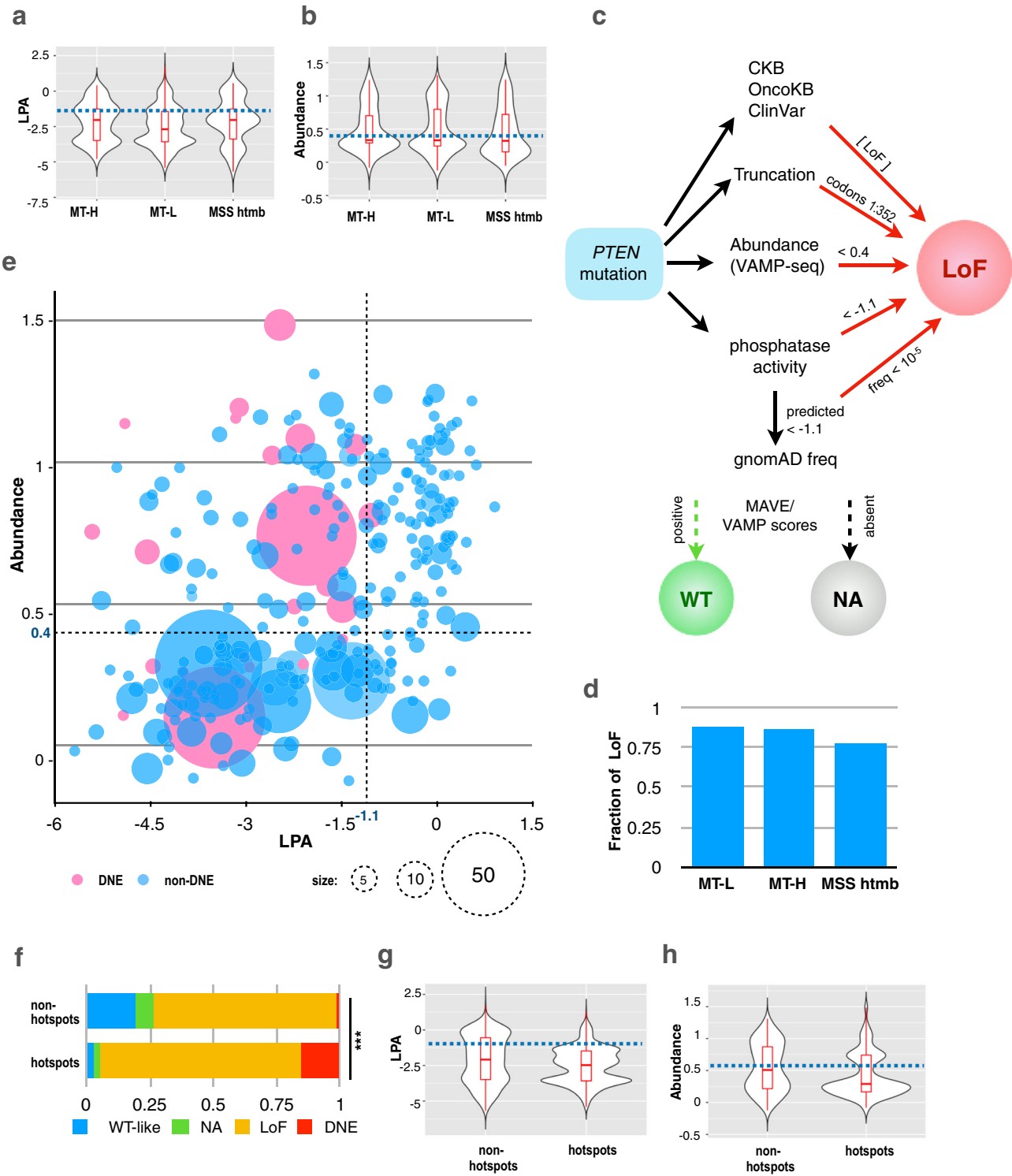

MMR deficiency (IDT and associated signatures, found in the MT-H subset) (Fig. 7d).

**The co-segregation pattern of *PTEN* mutations.** Common mutations associated with CRC pathogenesis inactivate *APC*, *TP53*, or *SMAD4*, or activate *RAS* (*KRAS* and *NRAS*), *RAF*, and *PI3K*[3,54]. Previous studies of the CRC cases have noted that mutations impairing or eliminating the function of PTEN co-occur with mutations in *KRAS*, *PIK3CA*, and *SMAD4*, but tend to be mutually exclusive with those inactivating *TP53*[55,56]. We

identified a similar pattern in the cumulative set of CRCs (Supplementary Table 27). The size of the cohort analyzed here allowed us to parse covariance between the CRC subclasses and to distinguish between segregation patterns in subsets with *PTEN* deletions versus point mutations.

Notably, the segregation pattern previously reported for CRC mutations largely reflected the pattern in MT-L tumors (Fig. 8a). Overall, *PTEN* alterations most commonly occurred in tumors bearing only *APC* mutations (14%), or with *APC* and *KRAS* mutations, and were least likely to co-occur in tumors bearing

**Fig. 6 LPA and abundance analysis of PTEN protein associated with mutations common in distinct tumor subtypes. a, b** Distribution of lipid phosphatase activity (LPA) (**a**) and abundance (VAMP-seq) (**b**) scores for MT-L, MT-H, and MSS-htmb tumors. LPA scores less than −1.10 (horizontal dashed line) are considered significantly impaired for phosphatase activity. VAMP-seq scores of 0.4 (horizontal dashed line) or less are considered significantly less abundant than wt protein. Box plots indicate median (middle line), 25th, 75th percentile (box), and 5th and 95th percentile (whiskers). Sample sizes and box plot parameters (low whisker, 25th percentile, median, 75th percentile, high whisker) for LPA are: MT-H, $n = 233$, boxplotstats = (−4.79; −3.49; −2.04; −1.26; 0.41); MT-L, $n = 915$, boxplotstats = (5.41; −3.58; −2.69; −1.43; 1.73); MSS htmb, $n = 195$, boxplotstats = (−5.69; −3.38; −2.04; −1.26; 0.56), sample sizes and box plot parameters (low whisker, 25th percentile, median, 75th percentile, high whisker) for abundance data are: MT-H, $n = 92$, boxplotstats = (−0.08; 0.29; 0.33; 0.70; 1.24); MT-L, $n = 441$, boxplotstats = (−0.12; 0.25; 0.33; 0.80; 1.31); MSS htmb, $n = 84$, boxplotstats = (−0.05; 0.16; 0.32; 0.73; 1.24), Exact $p$-values for the comparisons (using a Welch's unequal variances $t$-test and a Kolmogorov–Smirnov test) are provided in Supplementary Tables 21 and 22. **c.** Flowchart for dichotomization of variants into tentative loss of function (LoF) versus wild type-like (WT). See Materials and Methods for details. NA, information not available. **d**. Fraction of variants assigned as having some degree of LoF for MT-L, MT-H, and MSS-htmb tumors. Sample sizes: MT-H — 581; MT-L — 1319; MSS htmb — 203. **e** Combined LPA/abundance analysis for the complete CRC cohort. A pink color indicates dominant-negative variants, according to[53] and references therein. The size of the circle represents the number of samples for a given variant. **f–h** Distribution of mutation categories (**f**), lipid phosphatase activity (LPA) (**g**), and abundance (**h**) scores for the hotspot and non-hotspot subsets of PTEN mutations in the full CRC cohort. *** in (**f**), indicates $p$-value < 2.2e−16, as calculated using chi-squared contingency table test; Source Data are provided as a Source Data file. Dominant-negative mutations are significantly more common in the MT-L subset than in the MT-H subset, ~11% vs ~7.6% ($p$-value 0.0004), but the difference becomes insignificant if only point mutations are considered (12.4% versus 8.9%, $p$-value 0.24, calculated using the 2-sample test for equality of proportions with continuity correction). Box plots in (**g, h**) indicate median (middle line), 25th, 75th percentile (box), and 5th and 95th percentile (whiskers). *** indicates a $p$-value < 0.005, as calculated using a Kolmogorov–Smirnov test. Sample sizes: non-hotspots—764, hotspots—1360 (panels **f–h**); box plot parameters and exact $p$-values for the comparisons are provided in Supplementary Table 23.

*APC* and *TP53* mutations (4.7%) (Fig. 8b, c). In contrast, no co-occurrence with any of the tested genes was found in MT-H tumors, whereas in MSS-htmb tumors, *PTEN* mutations co-occurred with mutations in *APC* (Fig. 8a). More detailed analysis in the larger MT-L subset indicated similar co-occurrence and mutual exclusion patterns for both types of *PTEN* alterations with mutation of *SMAD4*, *KRAS*, and *TP53* (Fig. 8a). However, this pattern was only minimally significant for co-occurrence/mutual exclusion of *PTEN* deletions with *KRAS* and *TP53* mutations.

A primary function of PTEN is to oppose the activity of PI3K kinase in increasing PIP3 levels. As an alternative means of elevating PIP3, the catalytic subunit of PI3K, PI3KCA, is activated by mutation in a small but significant set of CRCs. Past studies have indicated a pattern of co-occurrence between *PTEN* mutations, and mutations activating PI3KCA[57,58], suggesting the interpretation that a subset of CRCs depends on PIP3 production to activate downstream signaling and that PTEN mutations are insufficient to produce adequate PIP3. Exploring this point in detail, we find that cumulatively, there is a strong co-occurrence of *PTEN* and *PIK3CA* mutations (Fig. 8d). This co-occurrence is driven by the MT-L and MSS-htmb sub-classes, but not observed in MT-H tumors (Fig. 8d). Interestingly, the co-occurrence of *PI3KCA* and *PTEN* missense/indel mutations trends lower with age in MT-L tumors (Fig. 8e). In contrast, there is highly significant mutual exclusion between *PTEN* deletions and *PI3KCA* mutations (Fig. 8d), sharply distinguishing this class of *PTEN* mutations from other classes. In further analysis (Fig. 8f), MT-L tumors with multiple *PTEN* mutations were more likely than those bearing a single mutation to have a co-occurring *PIK3CA* mutation (45% versus 25%, $p$-value 1.6e−05). This higher rate of co-occurrence in tumors with multiple PTEN mutations did not reflect a higher rate of overall mutation in those tumors, based on comparable TMB distributions in these subsets (Fig. 8f), raising the possibility that it identifies a class of CRC with particular dependence on AKT activity.

## Discussion
Because of the well-established roles of somatic *PTEN*-inactivating mutations as tumor-promoting, and of germline *PTEN* mutations as predisposing to multiple forms of cancer and other diseases, *PTEN* mutational patterns have long attracted much

interest[21,23,24,38]. In this context, this study makes several important contributions, particularly as *PTEN* mutations in CRC have been less studied, given the greater abundance of somatic PTEN mutations in other tumor types (including brain and endometrial), and the fact that germline mutations have a greater effect in elevating the risk for other cancer types (e.g., breast, renal, and thyroid)[59]. The data presented here provide an extensive list of *PTEN* mutational patterns in CRC overall, based on sufficient statistical power to separately analyze patterns of *PTEN* mutation in discrete CRC tumor subsets. The latter analysis identifies marked differences between the *PTEN* mutational profile observed in MT-L, MT-H, and MSS-htmb tumor classes, and in distinct tumor subsites, with some of these profiles associated with early versus late onset of CRC. The size of our cohort allowed us to confirm and extend earlier findings identifying the elevated association of PTEN mutations in MSI-H/MT-H tumors[23,24]. In contrast, although some differences in CRC mutational profile have been reported as distinguishing males and females, few differences were identified in this study[60]. Given the common use of drugs targeting EGFR and other receptor tyrosine kinases (RTKs) operating upstream of PI3K/PTEN in CRC[61,62], and the increasing exploration of drugs targeting PI3K and AKT in CRC tumors[63–65], a better understanding of *PTEN* mutational patterns is critical in predicting likely response to these therapeutic agents; the data provided here provide some suggestions into how distinct tumor classes will respond to these agents.

Integrated analysis of the effect of PTEN mutations on PTEN abundance and PTEN lipid phosphatase activity (Fig. 6) identifies notable differences between the properties of PTEN mutations occurring in MT-L versus MT-H tumors. In MT-H tumors, the prevalence of indels caused by IDT signatures resulted in a strong concordance between the damaging effect of mutations on lipid phosphatase activity and protein abundance, with the significant majority of mutations predicted to have a severe negative impact on both. In contrast, although there is some concordance of the effect of mutations on abundance and lipid phosphatase activity in MT-L tumors, this is less extensive, with some mutations affecting only lipid phosphatase activity, or only stability, and to intermediate degrees (Fig. 6). These mutations are likely to be pathogenic but have distinct properties. For example, given PTEN functions as a dimer, missense mutations that impair lipid phosphatase activity while maintaining protein stability and

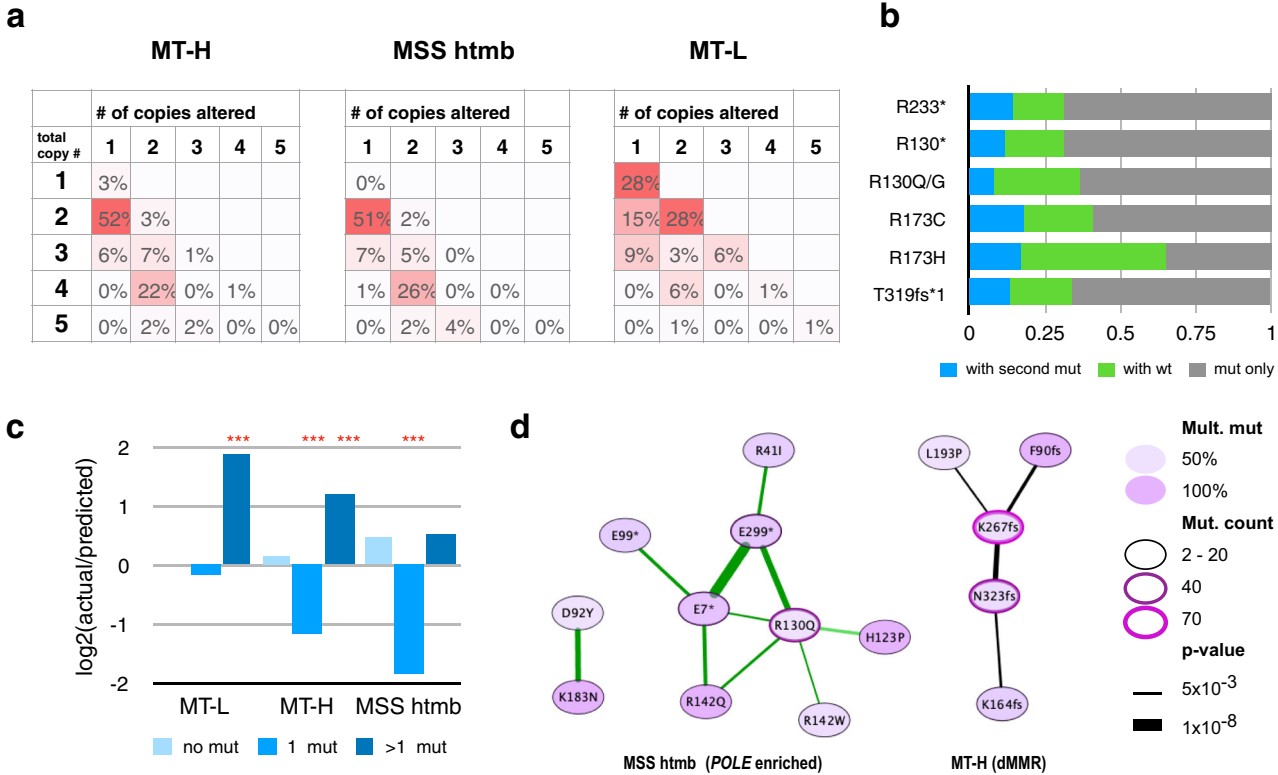

**Fig. 7 _PTEN_ mutation patterns and copy number alterations. a** Patterns of loss of heterozygosity (LOH) in MT-L, MT-H, and MSS-htmb tumors. The values shown indicate the frequency of co-occurrence of _PTEN_ mutations with altered copy number of _PTEN_ alleles. The vertical axis, the estimated total _PTEN_ copy number; a value of 1 indicates loss of one allele, while values of 3 or higher indicate increased gene copy number. The horizontal axis, the estimated copy number for the allele carrying a _PTEN_ mutation. Numbers in the cells indicate the percent of all mutations with a combination of total/ altered copy numbers, with more intense red shading emphasizing a greater abundance of the indicated combination of alleles. Sample sizes: MT-H, 601; MT-L,1332; MSS htmb, 202. **b** Occurrence of indicated hotspot mutations with wild type or additional mutated allele(s) ("with the second mut") in _PTEN_ for MT-L cohort. "Mut only", the only mutated allele is present. Sample sizes: R130*, 81; R130G/Q, 87; R173C, 60; R173H, 48; R233*, 92; T319fs, 60. **c** Skewed frequency of multiple _PTEN_ mutations. The actual frequencies of 0, 1, or >1 mutations in _PTEN_ were normalized to the frequencies expected based on a random distribution of mutation, and the log(2) of the resulting ratio was plotted. Zero on the vertical axis would correspond to a perfect match between predicted and actual frequencies; positive values indicate higher than predicted frequencies (with 1 corresponding to 2-fold), and negative values indicate the relative scarcity versus predicted numbers. Multiple mutations appear much more frequently than by chance in MT-L and MT-H subsets, while single mutations are much less frequent in the MT-H and MSS-htmb subsets. ***indicates _p_-value < 0.001, using a binomial distribution model; values for ratios plotted and exact _p_-values are provided in Supplementary Table 25. **d** Specific pairwise co-occurrences of PTEN hotspot mutations. Network visualization: Edge width reflects the degree of significance (−log10 of _p_-value, calculated using a binomial distribution model). The green edge indicates the presence of co-occurring _POLE_ mutations; most of these co-mutations involve the MSS-htmb cohort. A black edge indicates co-occurrence between the mutations compatible with signatures characteristic for MMR deficiency (dMMR; either IDT or SBS44). Mult. Mut—share of samples with a given mutation that co-occur with a second _PTEN_ mutation. Node color: darker color corresponds to a higher fraction of double mutation for a given mutation. Node border: increased width and shift towards purple color indicate a higher mutation count in the examined set.

capacity for dimerization are more likely to function as DNEs, eliminating the lipid phosphatase function of the residual wild type copy of PTEN. In addition, not all pathological _PTEN_ mutations affect lipid phosphatase activity, and mutations retain lipid phosphatase activity but resulting in an intact protein may target other important elements of PTEN function, including intracellular localization[66], protein phosphatase activity[67], non-catalytic scaffolding activity[68], or interaction with regulatory proteins[69]. This diversity, coupled with the fact that MT-L tumors are much more likely to have LOH for _PTEN_ that leaves only the mutated allele expressed (Fig. 7), suggests a more variable landscape of PTEN activity in MT-L versus MT-H tumors. Overall, the common feature of the not previously identified hotspot mutations was a property of reducing the lipid phosphatase activity of PTEN.

Interestingly, while there was no co-occurrence between _PTEN_ and _PIK3CA_ mutations in MT-H tumors, there was a clear pattern of co-segregation with mutations in MT-L tumors for all _PTEN_ mutation classes except complete deletion of _PTEN_ (Fig. 8), perhaps suggesting a greater selection for full activation of the PI3K pathway. The etiology of the _PIK3CA_ mutations in the various subsets of CRC tumors remains unclear, although it is interesting that one study has identified multiple mutations in MMR genes as associated with a high rate of _PIK3CA_ mutations in CRC[70]. The complicated pattern of association between _PIK3CA_ and _PTEN_ mutations identified here also suggests caution in evaluating clinical studies based on an assessment of PTEN protein; for instance, other work has identified mutual exclusion between _PIK3CA_ and _PTEN_ mutations in B cell lymphoma, in part based on immunohistochemical evaluation of PTEN protein expression - an approach biased to detect cases of _PTEN_ deletion (discussed in refs. [71,72]).

Loss of PTEN function has been implicated in resistance to single-agent PI3K inhibitors[73,74], necessitating the design of

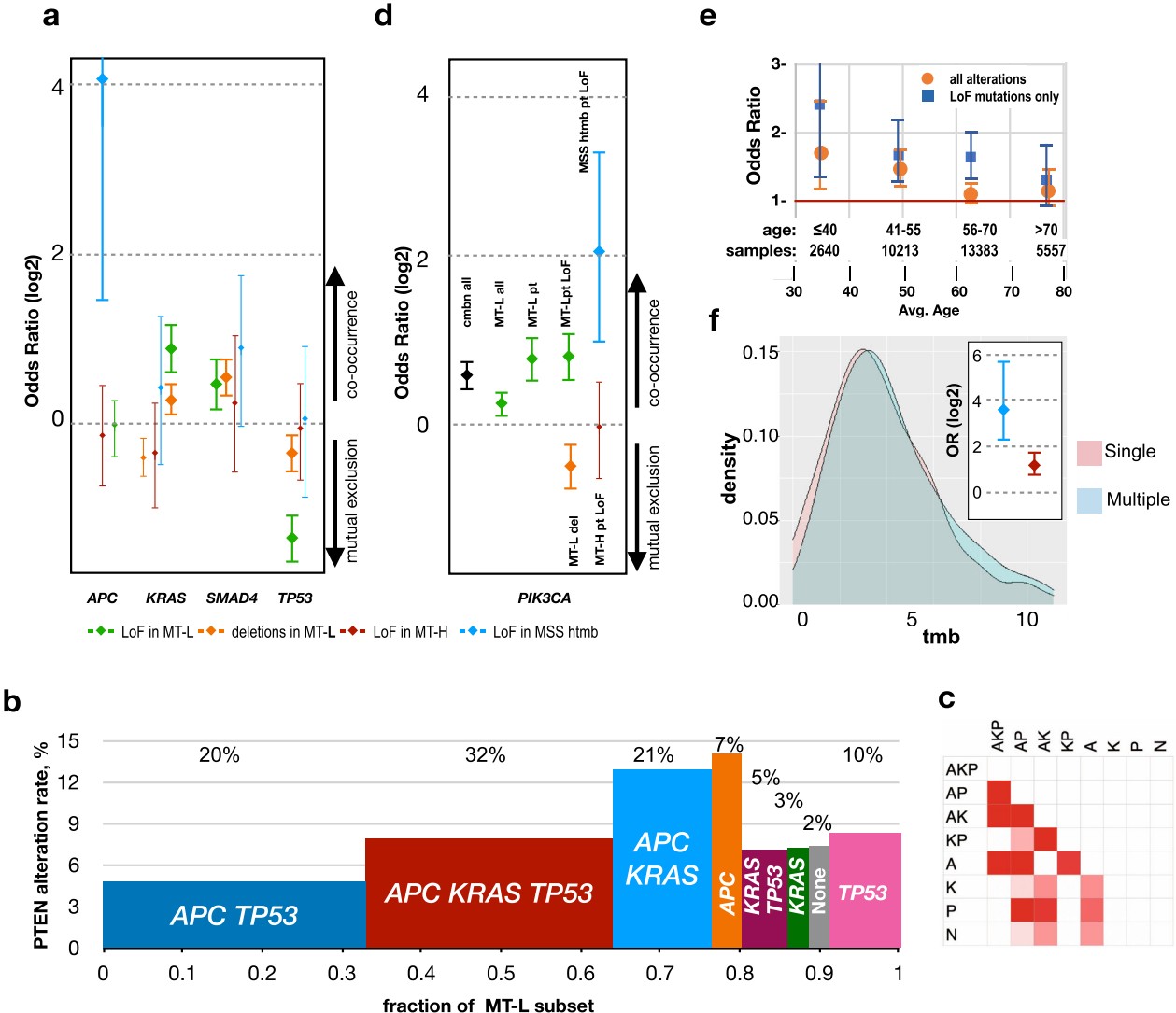

**Fig. 8 Co-occurrence patterns of *PTEN* mutations. a** Co-occurrence of LoF mutations or deletions in *PTEN* with any mutations in *TP53*, *KRAS*, *APC*, and *SMAD4*, in the MT-L, MT-H, and MSS-htmb cohorts. Co-occurrence is expressed as log2 of odds ratio, with the 95% confidence intervals shown (thicker bars indicate the result is statistically significant). Blue, PTEN LoF in MSS-htmb; red, PTEN LoF in MT-H; green, PTEN LoF in MT-L; orange, deletions in MT-L. Overall count of samples (panels **a**, **d**) bearing mutations in *APC*, 26910; *KRAS*, 17379; *SMAD4*, 7112; *TP53*, 26183; *PIK3CA*, 6665. Values for odds ratios plotted and exact *p*-values are provided in Supplementary Table 28. **b** Frequency of *PTEN* alteration in MT-L tumors containing mutations in A, *APC*; K, *KRAS*; P, *TP53*; N, none; in combinations as indicated. On the horizontal axis, the width of each column represents the fraction of MT-L tumors containing the indicated mutations in A, K, and/or P. For each group, the fraction of the overall *PTEN* alterations pool is indicated at the top. **c** Matrix of significance in *PTEN* alteration rate between the groups in panel (**b**); white, non-significant; pink to red, significant (FDR 0.05 to 10e−10). Sample sizes for groups (panels **b**, **c**): *APC KRAS TP53*, 9993, *APC TP53*, 10638; *APC KRAS*, 4027; *KRAS TP53*, 831; *APC*, 1177; *KRAS*, 850; *TP53*,2934; none, 783. **d** Co-occurrence of mutations in *PTEN* with mutations in *PI3KCA*, in subsets, as indicated. Cmbn all, all *PTEN* alterations in the analyzed set of CRC; MT-L all, all *PTEN* alterations in MT-L subset; MT-L pt, all *PTEN* mutations excluding copy number variations in the MT-L subset; MT-L pt LoF, same as preceding but only including *PTEN* mutations causing predicted loss of function; MT-L del, deletion of *PTEN*. Error bars indicate 95% confidence intervals. Values for odds ratios plotted and exact *p*-values are provided in Supplementary Table 28. **e** Co-occurrence of *PI3KCA* mutation with alterations in *PTEN*, as a function of age, in MT-L tumors. Orange, all alterations including deletions; blue, *PTEN* LoF mutations only. Data points with error bars (95% confidence intervals) crossing the horizontal axis line (OR = 1) are not statistically significant. **f** The TMB distribution for samples with single (pink) and multiple (blue) *PTEN* mutations; inset, co-occurrence of *PI3KCA* mutations with alterations in *PTEN*, as a function of the number of independent *PTEN* mutations in each sample (single, red, versus multiple, blue). Error bars represent 95% confidence intervals. Source Data are provided as a Source Data file.

combination therapies that block alternative routes of signaling[63,64]. For instance, the p110α-specific PI3K inhibitor alpelisib (BYL719) was found to effectively treat highly aggressive *BRAF*-mutated metastatic CRC when administered in combination with the RAF kinase inhibitor encorafenib and a monoclonal antibody targeting EGFR (cetuximab)[75]. Successful inhibition of the PI3K-AKT axis in

CRC and other tumors has been accomplished through dual PI3K/ mTOR inhibitor treatment[63,64]. Activation of WNT/β-catenin signaling, occurring in most CRC tumors, has been identified as a resistance mechanism to PI3K/mTOR treatment in CRC cell lines[76]; some data suggest that this resistance can be overcome by the addition of a MEK1/2 inhibitor, such as pimasertib[77]. However,

whether specific *PTEN* class and co-mutation patterns may impact PI3K-AKT pathway activity and ultimately treatment outcomes is not clear. These are important considerations in scenarios where a *PTEN* mutation may confer resistance to the clinically indicated therapy. For example, *PTEN* loss reduced the response of melanomas to immune checkpoint inhibitors[78] and gliomas to radiation therapy[79], which share similar mutations with CRC tumors. Interactions of *PIK3CA* mutations with response to other drugs have been observed[80]. Notably, tumors with both *PTEN* loss and activating mutations of *PIK3CA* are more resistant to cetuximab[81], further emphasizing the importance of considering mutation co-segregation patterns.

The analysis presented here cannot fully capture the impact of *PTEN* mutations, based on limitations in the dataset, which lacks prognostic or treatment information, and in some cases cannot exclude the mutations analyzed as somatic versus germline. Germline mutations in *PTEN* have been associated with some predisposition to CRC[15,82]; however, individuals with this syndrome are rare in the general population and this is not likely to represent a significant fraction of the assessed cohort. It is possible that there are some differences in mutational frequency or signature associated with the assessment of genes commonly included in panel testing for cancer, versus those captured in exome or whole-genome sequencing. For example, in this study average TMB values were 5.0 versus 3.6 identified for CRC MSS tumor in TCGA, and 53.7 versus 45.5 (TCGA) for MSI-H tumors (Supplementary Fig. 1). However, there are multiple potential reasons for higher TMB in the FMI data set, which may include larger cohort size; the tendency of FMI to sequence later stage tumors; or improvements in mutation technology reflected in the FMI cohort versus the older TCGA data set. Overall, we would expect differences affecting hotspots or signatures to be minor, but in the absence of a rigorous analysis of sufficiently large datasets in the existing literature, it is not possible to make a definitive statement. In addition, *PTEN* expression is also subject to epigenetic controls[83], which include promoter hypermethylation (particularly in MT-H tumors[84]) and targeting by microRNAs[85]; information bearing on the impact of these epigenetic control mechanisms is not available for the specimens analyzed here. However, based on the size of the dataset analyzed here, our study provides a detailed blueprint for segregating CRC tumors by *PTEN* mutation status within the landscape of various clinical subgroups and co-mutation patterns, providing context for subsequent analysis of epigenetic control of *PTEN* expression, and helping to enable rational design of future treatment combinations.

## Methods

**Comprehensive genomic profiling.** CGP was performed using the FoundationOne® or FoundationOne CDx assays (Foundation Medicine, Inc., Cambridge, MA, USA), as previously described in detail, on deidentified samples from patients who had been consented (but not compensated) for research. These specimens were collected from 2015 to 2019[86]. Patients have consented for sequencing from Foundation Medicine, Inc.; however, the need to obtain informed consent for our study was waived from the Western Institutional Review Board (Protocol No. 20152817), as the data were permanently de-identified before being provided to our group and could not be linked to individual patients. Typically, patients in the analyzed cohort had advanced or recurrent disease, or had recently failed treatment; data were not available for the time of initial diagnosis. The pathologic diagnosis of each case was confirmed by a review of hematoxylin- and eosin-stained slides and samples used for DNA extraction contained at least 20% tumor, with most specimens significantly exceeding this threshold, and passing stringent assessment for quality control. Hybridization capture of libraries prepared from exonic regions from a panel of cancer-related genes was applied to ≥50ng of DNA, sequenced to high, uniform median coverage (> 500 ×), and assessed for base substitutions, short insertions and deletions, copy number alterations, and gene fusions/rearrangements. Determination of the abundance of tumor DNA is taken into account when reporting copy number variants. Direct information was not available regarding germline mutation status for *PTEN* or other genes linked to hereditary risk of CRC (e.g., *MSH2*, *MLH1*, and others), as non-tumor DNA was not sequenced. However, in many although not all cases, evaluation of allele frequencies confirmed the *PTEN* mutations observed as somatic in origin. Comparison data sets for studies with information on *PTEN* mutation status, and sex, age, and tumor subsite were collected from the cBioPortal database (http://www.cbioportal.org/index.do).

**Statistical analysis.** Data were analyzed in R version 4.0.3 using RStudio. Relationships between mutations and patient characteristics were assessed using two-sided Fisher exact tests (including determination of the significant difference between the mutational spectra of dichotomized age, gender, or subsite groups, and multivariable logistic regression models). To allow for multiple mutations within a patient, the predictors used in these models were binary indicators for the presence/absence of particular mutations of interest. LPA and abundance profiles were compared using a *t*-test and a Kolmogorov–Smirnov test. To account for multiple comparisons of various types, we have lowered the threshold for statistical significance tenfold, to 0.005. Co-occurrence or mutual exclusion of mutations was calculated using Fisher's exact test.

**Identification of single-residue, 2D, and 3D hotspots.** Hotspot mutations and mutation-enriched stretches along the primary protein sequence were identified using previously described methods[22]. Briefly, to determine if a frequently mutated site on the *PTEN* protein constitutes a mutational hotspot, we have used a binomial distribution model with a *p*-value cutoff of 0.005. Similarly, to calculate whether non-hotspot mutations are enriched in certain linear stretches along the PTEN primary sequence, we used a sliding window of 5 aa and a binomial distribution model to identify larger regions of the primary structure that were, in sum, more commonly mutated than expected. 3D hotspots of missense mutations were calculated essentially as in ref. [87], with corrections to enhance reproducibility. To ensure confidence, missense mutations were calculated independently for three independent PTEN structures (PDB: 1D5R, 5BUG, and 5BZZ). Two residues with any pair of atoms within 5 Å were considered in contact if that distance was reproducible in at least 2 out of 3 structures, and within each structure, in at least 2 of 3 of the chains. A 3D cluster (defined by a central residue and at least one contact neighbor residue) was nominated as significantly mutated if the total number of mutations, combined across all its residues, was significantly higher than expected by chance, as determined by a permutation-based test, using a *p*-value cutoff of 0.005. For display of the distribution of mutations on the folded protein structure, figures were prepared using the program PyMOL[88], based on a PTEN structure deposited in the PDB (1D5R)[27].

**Assessment of PTEN variant functionality.** The likelihood that specific mutations impair one or more PTEN functions was derived from the integration of multiple sources, in addition to mutations explicitly characterized in detail in the scientific literature. Functional annotation for damaging *PTEN* mutations was collected from the Clinical Knowledgebase (CKB) ([48] https://ckb.jax.org, accessed 08.2019; "loss of function" or "loss of function—predicted"), OncoKB ([89]https://www.oncokb.org, accessed 11.2020; "oncogenic" and "likely oncogenic") and Clinvar ([49] https://www.ncbi.nlm.nih.gov/clinvar, accessed 12/2020; "pathogenic" or "likely pathogenic"). Mutations with varying assessments amongst these resources were considered damaging. All truncating mutations occurring in codons 1–352 were considered damaging. Splice mutations were provided in Human Genome Variation Society (HGVS) cDNA nomenclature[90] based on the reference transcript NM_000314. A chi-square test was used ($p < 0.05$) to identify significant differences in the fraction splice mutations between CRC subsets. Rare *PTEN* alterations, such as exon skipping, intron retention, rearrangements, and truncations, were identified by FMI as previously described[86].

In addition, estimates of lipid phosphatase activity and protein abundance were based on data reported in refs. [43,45], including subsequent reanalysis in refs. [43–45]. We used cut-offs of fitness score < −1.1 as indicating reduced phosphatase activity or a VAMP-seq score < 0.4 for substantially reduced abundance. For the predicted lipid phosphatase values only, an additional check for the population frequency was performed in GNOMAD v.2.1 ([91], https://gnomad.broadinstitute.org); all variants predicted to be function-impairing occurred at a frequency $<10^{-5}$ in the general population. The relationship between abundance and lipid phosphatase activity was established using Pearson's correlation, weighted to account for uneven sample sizes. For the purposes of Fig. 6e only, the abundance scores from[53] were used for the variants with no abundance score in ref. [43].

**Mutational signatures.** Information on mutational signatures (v 3.1) prevalent in CRC was downloaded from the COSMIC database (https://cancer.sanger.ac.uk/cosmic/signatures/[92]). Compatibility of the detected mutation with a given signature was determined by matching observed mutations to the most frequent nucleotide base changes, together with its trinucleotide context, in each signature. Where multiple signatures were compatible with the mutation in question, the signature most active in a specific biological context based on the scientific literature was used.

**Loss of heterozygosity (LOH).** LOH analysis was performed by comparing the copy numbers for total and mutated alleles, and sorting the samples into three

groups: those containing a mutated *PTEN* allele alone, those containing both a mutated *PTEN* allele and a wild-type *PTEN* allele, and those containing multiple mutated *PTEN* alleles. Samples, where only mutated PTEN allele was present (no wild type or additional mutations), were interpreted as LOH.

**Reporting summary**. Further information on research design is available in the Nature Research Reporting Summary linked to this article.

## Data availability

Consented data that can be released for publication are included in the article and its supplementary files and include permanently de-identified data on *PTEN* mutation status, the presence of mutations in other genes noted in the study, and sex, age, and tumor subsite for individuals profiled by FMI. Patients did not consent for the publication of underlying sequence data, nor can published data describe raw sequence data or link sequence data to patient clinical phenotypes. We sent a proposal describing the scope of our work through the Foundation Medicine website and we then filed out a study review form, which was checked by lawyers at each end. After the approval of a data transfer agreement, Foundation Medicine assigned us specialists in the dataset that were interested in colorectal cancer. Academic researchers can gain access to underlying Foundation Medicine data in this study by contacting Foundation Medicine using the coordinates on their website (https://www.foundationmedicine.com/contact), and filling out a data request form. Researchers and their institutions will be required to sign a data transfer agreement. The public web resources used in this paper are listed here: The cBioPortal for Cancer Genomics, https://www.cbioportal.org; AACR Project GENIE, https://genie.cbioportal.org; the Catalog Of Somatic Mutations In Cancer, https://cancer.sanger.ac.uk/cosmic; the Surveillance, Epidemiology, and End Results (SEER) Program, https://seer.cancer.gov. PyMol files for the visualization of hotspots on the PTEN structure (1D5R) and the Cytoscape file for the visualization of the co-occurrence between multiple *PTEN* mutations are provided with this paper as Supplementary Information Files (Supplementary Data 4.zip and Supplementary Data 5.zip). The remaining data are available within the Article, Supplementary Information, or Source Data file. Source data are provided with this paper.

## Code availability

The code used for identification of single-residue hotspots, mutation-enriched regions, and 3D hotspots is deposited at Github[93] and the corresponding DOI is as follows: https://doi.org/10.5281/zenodo.6149413. Source data are provided with this paper.

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

## Acknowledgements

We thank R. Dunbrack for guidance on protein structure analysis, and G. Romanov for the help with the computational prediction of methylation-dependent-nucleotide

substitutions. The authors were supported by NCI Core Grant P30 CA006927 (to Fox Chase Cancer Center), NIH R01 DK108195 (to EAG), R03 CA256234 (to IGS), Marie Skłodowska-Curie grant No. 896865 from the European Union's Horizon 2020 research and innovation program (to R.T.), and by Colon Cancer Alliance funding (to J.M.). The results reported here are in part based upon data generated by the TCGA Research Network: https://www.cancer.gov/tcga.

## Author contributions

J.N. and G.F. collected the data. I.G.S., J.N., V.P., R.T., M.I.P., G.A., and E.N. performed the data analysis and created figures and tables. E.A.G., I.G.S., and J.E.M. designed the study and wrote the paper.

## Competing interests

J.N. and G.F. are employed by FMI and own stock in Roche. The remaining authors declare no competing interests.
