## [Peer Review File · Nature Communications]

Comprehensive characterization of PTEN mutational profile in a series of 34,129 colorectal cancersReviewers' Comments:

Reviewer #1:

Remarks to the Author:

Serebriiskii and colleagues study a large cohort of individuals with colorectal cancer to identify specific patterns of somatic PTEN alterations associated with patient age, gross tumor location, MSS/MSI, and TMB.

1. In the title and especially in the Introduction, it may be important to clarify that the PTEN mutational profile is indeed somatic and not germline.
2. The authors seemed to have used the ages at the time of sequencing in their analyses. If my assumption is correct, why not use the age at CRC diagnosis which will give more accurate clinical and biological values?
3. Did the authors exclude patients with known germline mutations known to be associated with CRC (including PTEN)?
4. From the study, it was unclear whether an increased burden of PTEN alterations (ie, more than one hit) are associated with increased TMB?
5. In the Methods on page 18, the authors stated: "For the predicted lipid phosphatase values only, an additional check for the population frequency was performed in GNOMAD v.2.1 (<https://gnomad.broadinstitute.org>), and variants with the frequency $< 10^{-5}$ were considered as having some degree of loss of function." Can the authors please clarify how the frequency of a particular variant may be related to functional impact such as LoF?
6. The figures are generally very informative, but I found some to be missing appropriate visual annotations so that they cannot be understood without reading the legend. Can the authors add further colour figure legends to help clarify? (eg, Fig. 2G, Fig. 5D). Also, all gene instances would need to be italicised.
7. These current data's concept (though utilizing large sample size) is not new:

<https://pubmed.ncbi.nlm.nih.gov/11854177/>

<https://pubmed.ncbi.nlm.nih.gov/12163369/>

Please could the authors discuss their observations in the context of these prior publications?

Reviewer #2:

Remarks to the Author:

Comments for NCOMMS-21-24450 - Comprehensive characterization of PTEN mutational profile in a large cohort of colorectal cancer patients

The authors performed a genomic analysis on over 34k colorectal cancer patients derived from the FMI dataset in order to capture different mutation patterns in PTEN and assess their effect on protein function in 3 subgroup settings MT-L/MT-H/MSS-htmb similar to their published work from 2019. To support their findings, the authors utilize publicly available datasets such as TCGA for mutation frequency and VAMP and MAVS to assess protein abundance and activity associated with the defined mutations. However, despite their large cohort size and added value of defining different mutations effect on PTEN function in MT-L/MT-H/MSS-htmb sub groups the lack of a full statistical description,

mechanistic effect and missing figure legends make the manuscript hard to follow and to better assess the authors results.

Major comments:

The authors identified 3 different groups in their analysis - MT-L, MT-H and MSS htmB. MSI-H and MSS are mainly based on a clinical annotation or a stringent cutoff of 16 TMB. However, there are computational tools that could better assess the MSI status especially for borderline samples or samples lacking such an annotation. One such tool is referenced here <https://pubmed.ncbi.nlm.nih.gov/28892075/>. Moreover, subsampling of either WES or WGS data from the public datasets such as TCGA to the panel used to establish the correct mutation threshold would also be more appropriate. In addition, the MSS htmB should better be described as POLE enriched sub-group which would be more intuitive for the readers.

The authors use one of the largest CRC cohorts, however this dataset is created based on a panel of cancer genes rather than full WXS, therefore, shortcomings/ strengths should have been addressed in text or in the methods especially for mutational signature analysis. As an example - The authors state "Interestingly, although both the SBS1 and IDT signatures have been described as "clocklike" accumulating as a factor of age, an age-associated increase in these signatures among PTEN mutations was not observed in either the MT-L or MT-H cohorts". A possible explanation would be that due to a low number of mutations in the targeted panel they would not be powered to detect the increase over low frequency of mutations occurring by chance and would result in no/ low correlation. Statistical analysis performed - it would be beneficial if the authors would provide a more comprehensive method section detailing the use of each statistical test and where/why it was used, as well as their code for purposes of reproducibility. Moreover, the authors use a 0.005 as a more stringent P-value threshold to reduce false-positive results in such a large cohort size. However, using stringent cutoffs rather than adjusting for multiple hypotheses where appropriate and taking into account the number of tests applied would be a more suited approach. For example -in figure 8b they do use an FDR correction but in all others, they do not.

Do the authors take into account the sample's purity? Even with 500X sequencing depth, it would be worth testing the allelic fraction to see if purity has to be corrected as a covariate.

24 novel hotspots are stated to be identified by the authors but they are not discussed in detail for their effect.

In regards to the effect of mutations on the protein function and abundance, the authors should also consider comparing their results to additional proteomic available datasets such as the CPTAC consortium colon dataset with both proteomic and genomic data available or the CCLE dataset where multiple colon cancer cell lines are available. While those are much smaller cohorts the direct correlation between mutation and effect could be better addressed.

Figure 1a/1b - are almost identical to the authors' paper from 2019 and would need to be modified for purposes of novelty.

Minor comments -

Figures :

Inconsistency - Capital/lowercase letters for sub-panels (Ex. 3a), Most figures don't include a legend (Fig1c, Fig2- Fig8) or show the statistical significance either with p-value or an "*" annotation, correlation plots lack R and P values.

Figure 3A states in the text "schematic of PTEN protein domain structure, with the location of mutations" however the mutations are not marked. Also, the main protein domains discussed in the text are not marked in the figure (PDB/phosphatase / C2/ c-terminal) or on the bar plots below.

Figure 5B-C is not clearly described in the main text and figure 5D is a combined figure 3 lower panel.

Figure 6F - legend overlapping with figure

Figure 7 - add titles for subsets - POLE/ MMR

Figure 8B title should be added - MT-L tumors for clarity, Figure 8B/C - should contain the full name of mutated genes rather than a single letter which makes this figure confusing.

Reviewer #3:

Remarks to the Author:

The paper analyzes sequence data of a very large dataset of CRC patients permitting the identification of over 3,400 PTEN tumor suppressor mutations and allowing the analysis of PTEN mutation patterns in tumor subsets. Novel hotspot mutations were identified in the CRC cohort and differences in mutational profile are observed between MT-L, MT-H and MSS-htmb tumor classes. The study is interesting and has the potential for significant impact in cancer biology and clinical targeting of proteins upstream and downstream of PTEN in the context of CRC. Below are some points that can hopefully help further improve the manuscript.

-The authors state in the discussion that it is not clear whether specific PTEN class and co-mutation patterns may impact the PI3K/AKT pathway and ultimately treatment outcomes. It would therefore make a very nice addition to the paper, based on their data, to have some functional cellular data here on the PI3K/AKT pathway with a couple of hotspot missense mutations found in the different tumor subsets and assess response under steady-state conditions and in response to Cetuximab using a model system.

-Some of the figures need attention – e.g. Fig. 3a – K128 labeling is hidden behind the PTEN schematic; Figs 3c-f the mutation# given on the y-axis doesn't appear to correspond to the numbers given in brackets next to hotspot mutants; Fig. 6F legend covers x-axis etc.

-Page 16 – “Notably, tumors with both PTEN loss and activating mutations of PIK3CA are more sensitive to cetuximab” should be changed to “Notably, tumors with both PTEN loss and activating mutations of PIK3CA are more RESISTANT to cetuximab”

DETAILED REBUTTAL, REVIEWER COMMENTS

Reviewer #1, expert in colorectal cancer genetics/PTEN mutations (Remarks to the Author):

Serebriiskii and colleagues study a large cohort of individuals with colorectal cancer to identify specific patterns of somatic PTEN alterations associated with patient age, gross tumor location, MSS/MSI, and TMB.

Comment 1. In the title and especially in the Introduction, it may be important to clarify that the PTEN mutational profile is indeed somatic and not germline.

Response: We have not added the word “somatic” to the title (although we have to the introduction), because while the expectation and evidence supports the idea that the vast majority of mutations are somatic, due to the nature of the panel testing performed on the specimens, we cannot assign 100% of the mutations analyzed as somatic versus germline. This is discussed at length in response to comment 3, below; and we have added clarification to the methods, and final paragraph of the discussion.

Comment 2. The authors seemed to have used the ages at the time of sequencing in their analyses. If my assumption is correct, why not use the age at CRC diagnosis which will give more accurate clinical and biological values?

Response: This assumption is correct; but unfortunately, we cannot do this. This information is not typically available due to the variability in time lag between diagnosis and sequencing, based on how patients receive clinical care. For example, some samples are sequenced immediately upon diagnosis of an advanced stage of disease; others are sequenced upon treatment failure, or upon disease recurrence some time after initial diagnosis and treatment. Information of which patient belongs to which group is not captured by FMI. We have added a line to the methods to make this limitation clear.

Comment 3. Did the authors exclude patients with known germline mutations known to be associated with CRC (including PTEN)?

Response: Based on the testing performed, we did not have access to complete information about germline results for genes commonly associated with inherited risk for CRC (*MSH2*, *MLH1*, etc.), as we now state in the Methods. Hence, this response focuses on *PTEN*. Typically, accurate determination of whether mutations are germline or not requires parallel sequencing of tumor and non-tumor DNA (e.g. from peripheral blood cells). Given the main goal of FMI is to provide information to support clinical decisions, non-tumor DNA is not sequenced. In the absence of this information, in some cases it is possible to estimate germline versus somatic mutation frequency based on research-use-only (RUO) algorithms that compare frequency of mutations observed in tumor specimen. However, the effectiveness of these RUO algorithms is influenced by the proportion of tumor versus normal tissue in a pathological sample, and in the specimens analyzed in this study, the proportion of tumor varied over a broad range, with some specimens as low as ~20% tumor, and others higher than 90%. Therefore, while we have excluded from consideration in our study the seven *PTEN* mutations which were identified as germline with high confidence, in many cases the somatic-vs-germline status cannot be sufficiently determined. We have made these limitations clearer, both in the methods section, and in the discussion section, where we now discuss the potential impact of germline mutations on various incidences of mutations described in the study.

Comment 4. From the study, it was unclear whether an increased burden of PTEN alterations (i.e., more than one hit) are associated with increased TMB?

Response: There is an association, but the relationship is indirect. We have added more detailed data to support this conclusion to the results section. Briefly, our initially supplied data identified an overall trend of PTEN burden increasing with TMB, true for all cohorts (Fig. 2c). However, there was not a direct correlation between TMB and the likelihood of more than one PTEN hit. After normalizing PTEN burden to TMB, PTEN mutations did not occur more often in the hypermutated (MSS htmb) and MSI-H (MT-H) cohorts than in the MSS /low TMB cohort (Supplem. Fig. 2a). Co-occurrence of two PTEN mutations in the same sample was no higher than by chance in the hypermutated (MSS htmb) cohort. Notably, the odds ratio of two or more hits in PTEN occurring in the same sample was lower in the MT-H cohort (which has higher mutation burden) than in the MT-L cohort (Fig. 7c). We have also investigated whether subsequent mutations in PTEN would accrue more significantly with the increase of TMB than would single mutations, and again found no difference. We have extended the description of findings in the Results section of the study, and presented new data as Supplementary Figure 13.

Comment 5. In the Methods on page 18, the authors stated: “For the predicted lipid phosphatase values only, an additional check for the population frequency was performed in GNOMAD v.2.1 (<https://gnomad.broadinstitute.org>), and variants with the frequency $< 10^{-5}$ were considered as having some degree of loss of function.” Can the authors please clarify how the frequency of a particular variant may be related to functional impact such as LoF?

Response: We have corrected the methods to have a clearer phrasing regarding this point. To briefly explain the point, we hypothesized that given the importance of PTEN in development, a variant with a population frequency $> 10^{-5}$ and not annotated as associated with a developmental disorder was less likely to be loss of function, and would require more analysis than simply accepting predicted lipid phosphatase activities. Notably, none of the mutations we identified as function-impairing occurred at a frequency $> 10^{-5}$. The new phrasing is “For the predicted lipid phosphatase values only, an additional check for the population frequency was performed in GNOMAD v.2.1 (<https://gnomad.broadinstitute.org>); all variants predicted to be function-impairing occurred at a frequency $< 10^{-5}$ in the general population.”

Comment 6. The figures are generally very informative, but I found some to be missing appropriate visual annotations so that they cannot be understood without reading the legend. Can the authors add further colour figure legends to help clarify? (eg, Fig. 2G, Fig. 5D). Also, all gene instances would need to be italicised.

Response: We have added further color keys to the images noted, and hope we have appropriately italicized gene names in all cases.

Comment 7. These current data’s concept (though utilizing large sample size) is not new:

<https://pubmed.ncbi.nlm.nih.gov/11854177/>

<https://pubmed.ncbi.nlm.nih.gov/12163369/>

Please could the authors discuss their observations in the context of these prior publications?

Response: This has been done, with references added to the result and discussion sections; we thank the reviewer for pointing out these earlier studies. Unfortunately, journal length limits, which this article already exceeded, disallow more extensive discussion of these works.

Reviewer #2, expert in mutational signatures (Remarks to the Author):

Comments for NCOMMS-21-24450 - Comprehensive characterization of PTEN mutational profile in a large cohort of colorectal cancer patients

The authors performed a genomic analysis on over 34k colorectal cancer patients derived from the FMI dataset in order to capture different mutation patterns in PTEN and assess their effect on protein function in 3 subgroup settings MT-L/MT-H/MSS-htmb similar to their published work from 2019. To support their findings, the authors utilize publicly available datasets such as TCGA for mutation frequency and VAMP and MAVE to assess protein abundance and activity associated with the defined mutations. However, despite their large cohort size and added value of defining different mutations effect on PTEN function in MT-L/MT-H/MSS-htmb sub groups the lack of a full statistical description, mechanistic effect and missing figure legends make the manuscript hard to follow and to better assess the authors results.

Major comments:

Comment 1. The authors identified 3 different groups in their analysis - MT-L, MT-H and MSS htmb. MSI-H and MSS are mainly based on a clinical annotation or a stringent cutoff of 16 TMB. However, there are computational tools that could better assess the MSI status especially for borderline samples or samples lacking such an annotation. One such tool is referenced here <https://pubmed.ncbi.nlm.nih.gov/28892075/>.

Response: We have carefully read the reference the reviewer suggested, and also considered the list of methods provided in a very recent review on detection of microsatellite instability by Gilson et al., (PMID: 33804907). Unfortunately, most of the techniques listed in either study require either whole exome sequence (WES) analysis (as does the technique in the article the reviewer suggested), or the ability to analyze paired normal and tumor samples. FMI uses a comprehensive panel of cancer-related genes, but not WES; normal samples are also unavailable, unfortunately. We note, FMI now has a FDA-approved tool to determine MSS/MSI-H status, which includes calculation based on microsatellites. However, this tool is calibrated to the current version of the gene panel, and is not applicable to different, earlier versions of the panel; many of the CRC specimens we analyzed that lacked MS designation were obtained with earlier versions of the gene panel, not compatible with the current tool.

As alternative approaches, we have further probed the effect of applying more stringent cutoff values on the conclusions

Figure A. TMB distribution for MSS and MSI-H cohorts. More stringent TMB cutoffs are shown as dotted lines (corresponding to TMB 14 and 19). Samples with TMB<14 would be interpreted as MT-L (green) and a negligibly small fraction of MSI-H samples would be included. Samples with TMB>19 would be interpreted as MT-H (pink), and a small fraction of MSS samples would be included.

we make in this study (see Fig A). We have determined that if we alter the cut-offs to less than or equal to TMB 14, this results in sensitivity of 99% and specificity of 99.9% for MSS; and if we use a cut-off of $100 > \text{TMB} > 19$, this results in specificity and sensitivity in each case of 95% for MSI-H (Fig B); however, this improvement would come at the cost of having some samples with uncertain MS status. We have estimated, however, that the use of these most stringent cutoffs would alter the number of samples assigned to MSS by 14, which is 0.043% of the total (MT-L = MSS + low tmb) subset. The number of samples assigned to (MT-H = MSI-H + high tmb) subset would change by 7, or 0.44%. This would require recalculating essentially all the values in the paper, and would be unlikely to alter any conclusions.

To substantiate this later point, we have also estimated how much the results might change if we completely eliminated the samples with unknown MS status from the analysis. These samples currently comprise slightly less than 9% of our dataset. As a pilot experiment, we have recalculated two analyses: the fraction of samples with single and multiple PTEN alterations, and the age trends for PTEN alterations; we show here the age trend graph (Fig C), with nearly identical results as for the initial analysis. The maximum difference we see is 0.2% between the MT-L versus the MSS cohorts, and 4.3% between MT-H and MSI-H cohorts for the estimate of PTEN prevalence. This corresponds to a 2.8% in correlation coefficients for MT-L vs MSS cohorts for the age trend. We note, these differences lie well within the margin of error of calculations originally made. Based on these preliminary studies, we believe that further refinement of MS status prediction techniques for the 9% of samples in our dataset would slightly reduce statistical power in the trends we report, but not alter qualitative conclusions, while requiring recalculation and recreation of virtually all the Figures in the study.

We also note, the reviewer suggests using the approach outlined by Maruvka *et al.*, to calculate somatic changes in MS indels. Although MSMuTect, developed for the analysis of whole exome sequencing data, could be potentially adapted for the use of panel testing, the number of available indels from panel testing would be negligible compared to the Maruvka *et al.* publication, and hence unlikely to further improve

separation of MSS and MSI-H specimens.

Based on these issues, we have explicitly added commentary on the issue of mutation threshold calling uncertainty in the discussion section, final paragraph.

Comment 2. Moreover, subsampling of either WES or WGS data from the public datasets such as TCGA to the panel used to establish the correct mutation threshold would also be more appropriate.

Response: We interpret this comment to suggest that as a means of establishing TMB thresholds as predictive of microsatellite status, we take TCGA WES/WGS data, purge any mutations that occur in any gene beyond those present in FMI panel, apply tmb-predicting algorithm, and (using the known subsets of MSI-H vs MSS) establish the cutoff for separating one from another. There are a number of reasons why we think this would not improve the separation of classes in this study. The most important reflect an extensive published comparison of various techniques to establish tmb for tumor samples using subsampling (Yao et al 2020, PMID: 32188929), which demonstrated that each produces somewhat different results, and reflects our analysis of the TCGA dataset in response to this comment.

To this point, we collected TCGA data available for CRC, with MSI-H versus MSS annotation, and created a TMB profile, which we then compared to the TMB profile for the FMI dataset we analyzed (Fig D). For the known MSI-H datasets, TMB mean values were 45.5 for TCGA, and 53.7 for FMI (18% higher in FMI); for the MSS datasets, the mean values were 3.6 and 5.0 (40% higher in FMI). We cannot definitively explain why these values differ. The TMB-calling algorithm was developed at FMI on the basis of careful comparison of WGS, WES and panel sequencing, and is FDA-approved. A recent independent analysis (cited above, PMID: 32188929) subsampling TCGA WES indicated that results obtained from FMI panel are within a 95% confidence interval of true TMB. Nevertheless, as TCGA members and FMI use non-identical mutation calling approaches, the number of mutations set in a threshold would be predicted to be different. In addition, the FMI dataset is more biased to later stage tumors than the TCGA. It is possible that sequencing technology in detecting mutations is now more effective at detecting mutations at lower clonality than the data deposited in TCGA. Whatever the contributing factors, these differences and the fact that the size of the FMI training set in the present data set far exceeds the number of samples available in TCGA, support our current approach. Since the subset of our data in question only represents less than 9% of all datapoints, we believed it would be counterproductive to undertake this analysis, which would

require extensive efforts without providing clear benefits. We have added Fig D to the manuscript as a new panel (Supplementary Figure S1d), and also discussed this issue of differences between the TCGA and FMI assessments of TMB (legend to Fig 1).

Comment 3. In addition, the MSS htmb should better be described as *POLE* enriched subgroup which would be more intuitive for the readers.

Response: We have modified the text and the key on Fig 7e to more clearly indicate that the MSS-htmb subset is *POLE* enriched. However, not all of these specimens had *POLE* mutations, and in performing this analysis, we have only used TMB for dichotomization of MSS samples. Given the growing evidence that mutation of other genes (for instance, *POLD1*) can contribute to high TMB, and the fact that a number of the samples screened with earlier panels do not have information available, we prefer to keep the designation of “MSS htmb”, so as not to over-assign mechanism in the absence of clear data.

Comment 4. The authors use one of the largest CRC cohorts, however this dataset is created based on a panel of cancer genes rather than full WXS, therefore, shortcomings/ strengths should have been addressed in text or in the methods especially for mutational signature analysis. As an example - The authors state “Interestingly, although both the SBS1 and IDT signatures have been described as “clocklike” accumulating as a factor of age, an age-associated increase in these signatures among PTEN mutations was not observed in either the MT-L or MT-H cohorts”. A possible explanation would be that due to a low number of mutations in the targeted panel they would not be powered to detect the increase over low frequency of mutations occurring by chance and would result in no/ low correlation.

Response: To address this (and partially the next) comment, and to comply with the journal’s requirement to provide extended data for the Figures, we provide a new Supplementary Table S17 with the correlation coefficients and p-values for the SBS1 and IDT signatures, as well as with the number of datapoints used to calculate them; these analyses imply that the number of datapoints would be sufficient to detect a clock-like trend, if it existed. An interesting conjecture is that while clock-like signatures pertain at the genome level, the “clock” follows different rules for different genes, based on factors including degree of selection pressure, degree of gene expression, and other parameters. However, the reviewer’s point is well taken, and we have added commentary (final paragraph) about evaluating mutational trends from panel testing versus WES to the discussion.

Comment 5. Statistical analysis performed - it would be beneficial if the authors would provide a more comprehensive method section detailing the use of each statistical test and where/why it was used, as well as their code for purposes of reproducibility. Moreover, the authors use a 0.005 as a more stringent P-value threshold to reduce false-positive results in such a large cohort size. However, using stringent cutoffs rather than adjusting for multiple hypotheses where appropriate and taking into account the number of tests applied would be a more suited approach. For example -in figure 8b they do use an FDR correction, but in all others they do not.

Response: In Figs. 8b, c in the current study, we had a defined set of $8 \times 8 = 64$ comparisons, so we felt it was appropriate to apply FDR. However, in our prior Nat Comm manuscript on RAS mutations in CRC, the reviewers specifically and correctly pointed out that we had a large number of very diverse tests (hotspot calculation, age trends, co-occurrence etc) so it is not clear how to get a total “count of tests applied”. Guided by this past critique (which was accompanied by a direct suggestion from the earlier Nat Comm reviewers), and taking into

account the similarities between our past and current studies in that most of our analyses had only one or two comparisons, we did not apply FDRs for most of them. Instead, we chose to set a much stricter threshold.

In addition, we have added 17 supplementary tables with underlying data points and/or the parameters pertinent to the evaluation of statistical comparisons in the main and supplementary figures (e.g., sample sizes in each analysis, regression coefficients, p-values, quartile data for density plots, and other relevant calculations). We have also added a statement to the manuscript that the code used for analysis is available on request.

Comment 6. Do the authors take into account the sample's purity? Even with 500X sequencing depth, it would be worth testing the allelic fraction to see if purity has to be corrected as a covariate.

Response: FMI panel testing is based on specimens with a minimum threshold of 20% tumor, with most samples significantly exceeding this threshold, as determined by a pathologist. Determination of the abundance of tumor DNA is taken into account when reporting copy number variants, as part of the FMI processing pipeline, and only samples which passed stringent thresholds for quality were included in our dataset. This information has been added to the Methods section.

Comment 7. 24 novel hotspots are stated to be identified by the authors but they are not discussed in detail for their effect.

Response: In an initial version of this manuscript, we included much more description of the hotspots, but most of this text was eliminated, along with associated references, to try to comply with text and reference limits for the journal. The submitted manuscript is still over the recommended length limit for Nat Comm. We have now included text noting that almost all of the novel hotspots result in the loss of LPA or/and abundance. Information on loss of LPA or/and abundance, as well as for annotations regarding these mutations in clinically relevant databases, is available in Supplementary Data file 2. Unfortunately, it is impossible to insert full discussion for this large number of hotspots, given that this manuscript exceeds length limits for the journal.

Comment 8. In regards to the effect of mutations on the protein function and abundance, the authors should also consider comparing their results to additional proteomic available datasets such as the CPTAC consortium colon dataset with both proteomic and genomic data available or the CCLE dataset where multiple colon cancer cell lines are available. While those are much smaller cohorts the direct correlation between mutation and effect could be better addressed.

Response: We have attempted to address this point by retrieving PTEN protein abundance scores for the TCGA CRC (pan-cancer 2018) cohort, using both CPTAC and RPPA datasets (separately). Only 28 samples in these datasets have mutations in *PTEN*. Among these, 13 samples have double mutations, complicating interpretation. Of the remaining 15 samples, none has mass spectrometry data (in general, only ~100 CRC samples have been profiled), and only 10 have RPPA data.

We have also accessed the most recent cell line data through the DepMap portal (<https://depmap.org>). Only 25 CRC cell lines have proteomics data (3 of them have *PTEN* mutations), and 56 have RPPA data, of which 9 have *PTEN* mutations; however, copy number differences for *PTEN* (up to 4 copies) are also reported among them. Despite the promise that

proteomics holds for the suggested analysis, the current datasets are too small to draw meaningful conclusions, especially if keeping in mind that *PTEN* is a subject of regulation on additional levels, including promoter methylation and posttranslational modifications (PMID: 30738865, cited in the mss), which all have to also be taken into account. While we wish we could address this suggestion, which we agree would add to the study, the data we have been able to access do not allow it.

Comment 9. Figure 1a/1b - are almost identical to the authors' paper from 2019 and would need to be modified for purposes of novelty.

Response: While Figure 1a has a similar appearance as that in the 2019 study on RAS proteins, it is an analysis of a much larger dataset (~34K versus ~ 14K specimens) and is the most effective way we can envision to convey the relevant information, so we have retained it. We have redrawn Figure 1b to achieve a different graphic look.

Minor comments

Comment. Figures - Inconsistency - Capital/lowercase letters for sub-panels (Ex. 3a), Most figures don't include a legend (Fig 1c, Fig 2- Fig 8) or show the statistical significance either with p-value or an "*" annotation, correlation plots lack R and P values.

Response: We have calculated and added R and p values, and made the other requested changes regarding display of statistical information (as well as provided it in additional Supplementary Tables, as outlined in our response to comment 5). We apologize for inadvertent omissions in the originally submitted manuscript. We are puzzled by the comment about absent legends; these were present for all figures, at the end of the manuscript text. Hence, we surmise the reviewer means a key on the image itself, and have made additions to many panels throughout all Figures where we thought they would improve comprehensibility.

Comment. Figure 3A states in the text "schematic of PTEN protein domain structure, with the location of mutations" however the mutations are not marked. Also, the main protein domains discussed in the text are not marked in the figure (PDB/phosphatase / C2/ c-terminal) or on the bar plots below.

Response: We thank reviewer for catching this. We have added color legends indicating "the main protein domains discussed in the text (PDB/phosphatase / C2/ C-terminal)" at the bottom of Figures 3, 4, S3, S5, and S9.

Comment. Figure 5B-C is not clearly described in the main text and figure 5D is a combined figure 3 lower panel.

Response: We have improved the clarity of description of Figs 5b and 5c in the text. We agreed with the reviewer that Fig 5d was redundant with earlier material, and has now been removed.

Comment. Figure 6F - legend overlapping with figure

Response: This has now been corrected.

Comment. Figure 7 - add titles for subsets - POLE/ MMR

Response: This has now been done.

Comment. Figure 8B title should be added - MT-L tumors for clarity,

Response: This has been done.

Comment. Figure 8B/C - should contain the full name of mutated genes rather than a single letter which makes this figure confusing.

Response: We have improved the visual clarity of the figure, and we have replaced single letter abbreviations with the full name of mutated genes in the Figure 8b. For Figure 8c, not using single letter code was incompatible with the graphic composition used; however, we have improved the written legend for the figure.

Reviewer #3, expert in PTEN and cancer (Remarks to the Author):

The paper analyzes sequence data of a very large dataset of CRC patients permitting the identification of over 3,400 PTEN tumor suppressor mutations and allowing the analysis of PTEN mutation patterns in tumor subsets. Novel hotspot mutations were identified in the CRC cohort and differences in mutational profile are observed between MT-L, MT-H and MSS-htmb tumor classes. The study is interesting and has the potential for significant impact in cancer biology and clinical targeting of proteins upstream and downstream of PTEN in the context of CRC. Below are some points that can hopefully help further improve the manuscript.

Comment. The authors state in the discussion that it is not clear whether specific PTEN class and co-mutation patterns may impact the PI3K/AKT pathway and ultimately treatment outcomes. It would therefore make a very nice addition to the paper, based on their data, to have some functional cellular data here on the PI3K/AKT pathway with a couple of hotspot missense mutations found in the different tumor subsets and assess response under steady-state conditions and in response to Cetuximab using a model system.

Response: We agree with the reviewer that *in vivo* analysis would be of interest, but given the complexity of the data presented in this study, and the number of different experiments we would need to perform to generate meaningful conclusions, this would greatly alter the shape of the current study. Our hope is to leverage the insights developed in this manuscript to guide such future studies.

Comment. Some of the figures need attention – e.g. Fig. 3a – K128 labeling is hidden behind the PTEN schematic; Figs 3c-f the mutation# given on the y-axis doesn't appear to correspond to the numbers given in brackets next to hotspot mutants; Fig. 6F legend covers x-axis etc.

Response: We have corrected these and other minor technical issues with our Figures, and thank the reviewer for pointing out these small errors.

Comment. Page 16 – “Notably, tumors with both PTEN loss and activating mutations of PIK3CA are more sensitive to cetuximab” should be changed to “Notably, tumors with both PTEN loss and activating mutations of PIK3CA are more RESISTANT to cetuximab”

Response: This has been done, and we apologize for the error.

Reviewers' Comments:

Reviewer #1:

Remarks to the Author:

The authors have addressed most of my critiques. Two main comments remain that must be reflected in a revised manuscript:

1. The title is still misleading. In the human genetics arena, this may automatically be interpreted as scanning a large cohort of patients for germline variants but they are looking at somatic variants. If the authors are avoiding adding 'somatic' to their title, then would recommend to distinguish between sequencing tumors versus patients – the authors are reporting on the tumours' PTEN variants, and not the patients' (eg, Comprehensive characterization of PTEN mutational profile in a large series of colorectal cancers).

2. In their Discussion (page 17, line 494), the authors state: "Germline mutations in PTEN have been associated with some predisposition to CRC, particularly in Lynch syndrome; however, individuals with this syndrome are rare in the general population and this is not likely to represent a significant fraction of the assessed cohort." This is factually incorrect – people who have germline PTEN mutations have PTEN hamartoma tumour syndrome (PHTS), which is different than Lynch syndrome (even if both have CRC as a component malignancy, each has its own underlying genetic aetiology), whereby the latter is caused by germline mutations in mismatch repair genes (eg, MSH2, MLH1, MSH6).

3. An incorrect reference from Charis Eng's group was also used. The study linking CRC to PHTS also emanates from her group but is Tan MH et al. Clin Cancer Res 2012 Jan 15;18(2):400-7, doi: 10.1158/1078-0432.CCR-11-2283 and Heald B et al. Gastroenterology 2010; 139:1927-33.

Reviewer #2:

Remarks to the Author:

The authors have satisfactorily responded to the reviewers' comments including text and figure edits as well as additional supporting analyses.

Reviewer #3:

Remarks to the Author:

The authors have sufficiently answered all of the reviewers' concerns and I believe it is ready for publication

Response to Reviewers.

Reviewer 1.

The authors have addressed most of my critiques. Two main comments remain that must be reflected in a revised manuscript:

Point 1. The title is still misleading. In the human genetics arena, this may automatically be interpreted as scanning a large cohort of patients for germline variants but they are looking at somatic variants. If the authors are avoiding adding 'somatic' to their title, then would recommend to distinguish between sequencing tumors versus patients – the authors are reporting on the tumours' PTEN variants, and not the patients' (eg, Comprehensive characterization of PTEN mutational profile in a large series of colorectal cancers).

Response. We have made the adjustment as indicated. Also, we have removed the word "large" to comply with Nature Communications style preferences. The current title reads, "Comprehensive characterization of *PTEN* mutational profile in a series of 34,129 colorectal cancers".

Point 2. In their Discussion (page 17, line 494), the authors state: "Germline mutations in PTEN have been associated with some predisposition to CRC, particularly in Lynch syndrome; however, individuals with this syndrome are rare in the general population and this is not likely to represent a significant fraction of the assessed cohort." This is factually incorrect – people who have germline PTEN mutations have PTEN hamartoma tumour syndrome (PHTS), which is different than Lynch syndrome (even if both have CRC as a component malignancy, each has its own underlying genetic aetiology), whereby the latter is caused by germline mutations in mismatch repair genes (eg, MSH2, MLH1, MSH6).

Response. We thank the reviewer for pointing out this error. We have removed the reference to Lynch syndrome.

Point 3. An incorrect reference from Charis Eng's group was also used. The study linking CRC to PHTS also emanates from her group but is Tan MH et al. Clin Cancer Res 2012 Jan 15;18(2):400-7, doi: 10.1158/1078-0432.CCR-11-2283 and Heald B et al. Gastroenterology 2010; 139:1927-33.

Response. We originally had included the Tan reference; we have now additionally included the Heald *at al* reference, and apologize for its initial omission.